# ON COVARIATE SHIFT OF LATENT CONFOUNDERS IN IMITATION AND REINFORCEMENT LEARNING

**Guy Tennenholtz**
Nvidia Research
Technion

**Assaf Hallak**
Nvidia Research

**Gal Dalal**
Nvidia Research

**Shie Mannor**
Nvidia Research
Technion

**Gal Chechik**
Nvidia Research

**Uri Shalit**
Technion

## ABSTRACT

We consider the problem of using expert data with unobserved confounders for imitation and reinforcement learning. We begin by defining the problem of learning from confounded expert data in a contextual MDP setup. We analyze the limitations of learning from such data with and without external reward, and propose an adjustment of standard imitation learning algorithms to fit this setup. We then discuss the problem of distribution shift between the expert data and the online environment when the data is only partially observable. We prove possibility and impossibility results for imitation learning under arbitrary distribution shift of the missing covariates. When additional external reward is provided, we propose a sampling procedure that addresses the unknown shift and prove convergence to an optimal solution. Finally, we validate our claims empirically on challenging assistive healthcare and recommender system simulation tasks.

## 1 INTRODUCTION

Reinforcement Learning (RL) is increasingly used across many fields to create agents that learn via interaction and reward feedback. Imitation Learning (IL, Hussein et al. (2017)) is concerned with learning via expert demonstrations without access to a reward function. Similarly, RL settings often use expert data to boost performance, eliminating the need to learn from scratch. In this work we consider the IL and RL paradigms in the presence of partially observable expert data.

While expert demonstration data is useful, in many realistic settings such data may be prone to hidden confounding (Gottesman et al., 2019), i.e., there may be features used by the expert which are not observed by the learning agent. This can occur due to, e.g., privacy constraints, continually changing features in ongoing production pipelines, or when not all information available to the human expert was recorded. As we show in our work, covariate shift of unobserved factors between the expert data and the real world may lead to significant negative impact on performance, frequently rendering the data useless for imitation (see Figure 1 and Theorem 2).

In this paper we define the tasks of imitation and reinforcement learning using expert data with unobserved confounders and possible covariate shift. We focus on a contextual MDP setting, where a *context* is sampled at every episode from some distribution, affecting both the reward and the transition between states. We assume that the agent has access to additional expert data, generated by an optimal policy, for which the sampled context is missing, yet *is observed* in the online environment.

We begin by analyzing the imitation-learning problem, (i.e., without access to reward) in Section 3. Under no covariate shift in the unobserved context, we characterize a sufficient and necessary set of optimal policies. In contrast, we prove that in the presence of a covariate shift, if the true reward depends on the context, then the imitation-learning problem is non-identifiable and prone to catastrophic errors (see Section 3.2 and Theorem 2). We further analyze the RL setting (i.e., with access to reward and confounded expert data) in Section 4. Figure 1 depicts a possible failure case of using confounded expert data with unknown covariate shift in a dressing task. Unlike the imitation setting,

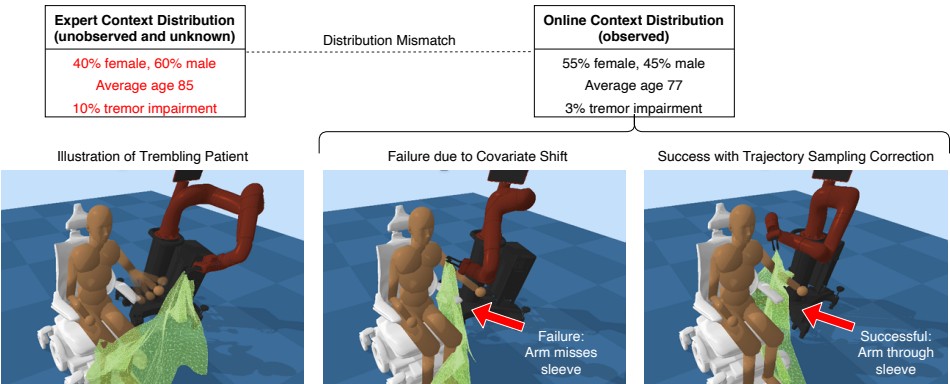

Figure 1: Failure of using confounded expert data under context distribution mismatch between online environment and expert data. Caregiver does not learn to perform well in a dressing task when covariate shift of hidden confounders is present but not accounted for.

we show that optimality can still be achieved in the RL setting while using confounded expert data with arbitrary covariate shift. We use a corrective data sampling procedure and prove convergence to an optimal policy.

Our contributions are as follows. (1) We introduce IL and RL with hidden confounding and prove fundamental characteristics w.r.t. covariate shift and the feasibility of imitation. (2) In the RL setting, under arbitrary covariate shift, we provide a novel algorithm with convergence guarantees which uses a corrective sampling technique to account for the unknown context distribution in the expert data. (3) Finally, we conduct extensive experiments on recommender system (Ie et al., 2019) and assistive-healthcare (Erickson et al., 2020) environments, demonstrating our theoretical results, and suggesting that confounded expert data can be used in a controlled manner to improve the efficiency and performance of RL agents.

## 2 PRELIMINARIES

**Online Environment.** We consider a contextual MDP (Hallak et al., 2015) defined by the tuple $\mathcal{M} = (\mathcal{S}, \mathcal{X}, \mathcal{A}, P, r, \rho_o, \nu, \gamma)$, where $\mathcal{S}$ is the state space, $\mathcal{X}$ is the context space, $\mathcal{A}$ is the action space, $P : \mathcal{S} \times \mathcal{S} \times \mathcal{A} \times X \mapsto [0, 1]$ is the context dependent transition kernel, $r : \mathcal{S} \times \mathcal{A} \times X \mapsto [0, 1]$ is the context dependent reward function, and $\gamma \in (0, 1)$ is the discount factor. We assume an initial distribution over contexts $\rho_o : \mathcal{X} \mapsto [0, 1]$ and an initial state distribution $\nu : \mathcal{S} \times \mathcal{X} \mapsto [0, 1]$.

The environment initializes at some context $x \sim \rho_o(\cdot)$, and state $s_0 \sim \nu(\cdot|x)$. At time $t$, the environment is at state $s_t \in \mathcal{S}$ and an agent selects an action $a_t \in \mathcal{A}$. The agent receives a reward $r_t = r(s_t, a_t, x)$ and the environment then transitions to state $s_{t+1} \sim P(\cdot|s_t, a_t, x)$.

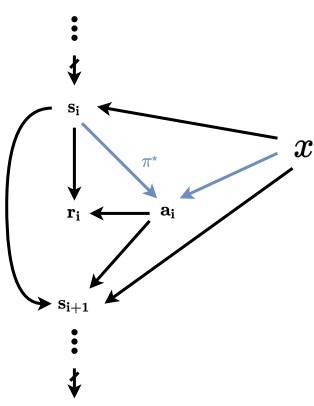

Figure 2: Causal Diagram Contextual MDP

We define a Markovian stationary policy $\pi$ as a mapping $\pi : \mathcal{S} \times \mathcal{X} \times \mathcal{A} \mapsto [0, 1]$, such that $\pi(\cdot|s, x)$ is the action sampling probability. We define the value of a policy $\pi$ by $v_{\mathcal{M}}(\pi) = \mathbb{E}_\pi \big[ (1 - \gamma) \sum_{t=0}^\infty \gamma^t r(s_t, a_t, x) \mid x \sim \rho_o, s_0 \sim \nu(\cdot \mid x) \big]$, where $\mathbb{E}_\pi$ denotes the expectation induced by the policy $\pi$. We denote by $\Pi$ the set of all Markovian policies and $\Pi_{\text{det}}$ the set of deterministic Markovian policies. We define the optimal value and policy by $v_{\mathcal{M}}^* = \max_{\pi \in \Pi} v_{\mathcal{M}}(\pi)$, and $\pi_{\mathcal{M}}^* \in \arg\max_{\pi \in \Pi} v_{\mathcal{M}}(\pi)$, respectively. Whenever appropriate, we simplify notation and write $v^*, \pi^*$. We use $\Pi_{\mathcal{M}}^*$ to denote the set of optimal policies in $\mathcal{M}$, i.e., $\Pi_{\mathcal{M}}^* = \arg\max_{\pi \in \Pi} v_{\mathcal{M}}(\pi)$. We also define the set of catastrophic policies $\Pi_{\mathcal{M}}^\dagger$ as the set

$$\Pi_{\mathcal{M}}^\dagger = \arg\min_{\pi \in \Pi} v_{\mathcal{M}}(\pi). \tag{1}$$

We later use this set to show impossibility of imitation under arbitrary covariate shift and a context-independent transition function.

**Expert Data with Unobserved Confounders.** We assume additional access to a confounded dataset consisting of expert trajectories $\mathcal{D}^* = \left\{ (s_0^i, a_0^i, s_1^i, a_1^i, \ldots, s_H^i, a_H^i) \right\}_{i=1}^n$, where $a_j^i \sim \pi^* \in \Pi_\mathcal{M}^*$. The trajectories in the dataset were sampled i.i.d. from the marginalized expert distribution (under possible context covariate shift) $P^*(s_0, a_0, s_1, a_1, \ldots, s_H) = \sum_x \rho_e(x)\nu(s_0|x) \prod_{t=0}^{H-1} P(s_{t+1}|s_t, a_t, x)\pi^*(a_t|s_t, x)$, where $\rho_e$ is some distribution over contexts. Importantly, $\rho_e$ does not necessarily equal $\rho_o$ – the distribution of contexts in the online environment. Notice that it is assumed that $\pi^*$ that generated the data had access to the context $x^i$ (i.e., $\pi^*$ is context-dependent), though it is missing in the data.

In this work, we consider two settings:

1. **Confounded Imitation Learning** (Section 3): The agent has access to confounded expert data (with context distribution $\rho_e$) as well as real environment $(\mathcal{S}, \mathcal{X}, \mathcal{A}, P, \rho_o, \nu, \gamma)$, *without* access to reward.

2. **Reinforcement Learning with Confounded Expert Data** (Section 4): The agent has access to confounded expert data (with context distribution $\rho_e$) as well as real environment $\mathcal{M} = (\mathcal{S}, \mathcal{X}, \mathcal{A}, P, r, \rho_o, \nu, \gamma)$, *with* access to reward.

In both settings we aim to find a context-dependent policy which maximizes the cumulative reward. The confounding factor here is w.r.t. the unobserved context and distribution $\rho_e$ in the offline data.

**Marginalized State Action Distribution.** We denote the state-action frequency of a policy $\pi \in \Pi$ given context $x \in \mathcal{X}$ by $d^\pi(s, a|x) = (1 - \gamma) \sum_{t=0}^\infty \gamma^t P^\pi(s_t = s, a_t = a|x, s_0 \sim \nu(\cdot|x))$, where $P^\pi$ denotes the probability measure induced by $\pi$. Similarly, given a distribution over contexts, we define the marginalized state-action frequency of a policy $\pi$ under the corresponding context distribution by

$$d_{\rho_o}^\pi(s, a) = \mathbb{E}_{x \sim \rho_o}[d^\pi(s, a \mid x)] \qquad \text{(online environment)},$$

$$d_{\rho_e}^{\pi^*}(s, a) = \mathbb{E}_{x \sim \rho_e}\left[d^{\pi^*}(s, a \mid x)\right] \quad \text{(offline expert data)}.$$

**A Causal Perspective.** Our work sits at an intersection between the fields of RL and Causal Inference (CI). We believe it is essential to bridge the gap between these two fields, and include an interpretation of our model using CI terminology in Appendix D, where we equivalently define our objective as an intervention over the unknown distribution $\rho_e$ in a specified Structural Causal Model, as depicted in Figure 2.

## 3 IMITATION LEARNING WITH UNOBSERVED CONFOUNDERS

In this section, we analyze the problem of confounded imitation learning, namely, learning from expert trajectories with hidden confounders and without reward. Similar to previous work, we consider the task of imitation learning from expert data in the setting where the agent is allowed to interact with the environment (Ho & Ermon, 2016; Fu et al., 2017; Kostrikov et al., 2019; Brantley et al., 2019). In the first part of this section we assume no covariate shift between the online environment and the data is present, i.e., $\rho_e = \rho_o$. We lift this assumption in the second part, where we focus on the covariate shift of the hidden confounders.

### 3.1 NO COVARIATE SHIFT: $\rho_o = \rho_e$

We first consider the scenario in which no covariate shift is present between the offline data and the online environment, i.e., $\rho_o = \rho_e$. We begin by defining the marginalized ambiguity set, a central component of our work.

**Definition 1.** *For $\pi \in \Pi$, we define the set of deterministic policies that match the marginalized state-action frequency of $\pi$ by* $\Upsilon_\pi = \left\{ \pi' \in \Pi_{det} : d_{\rho_o}^{\pi'}(s, a) = d_{\rho_e}^\pi(s, a) \quad \forall s \in \mathcal{S}, a \in \mathcal{A} \right\}$.

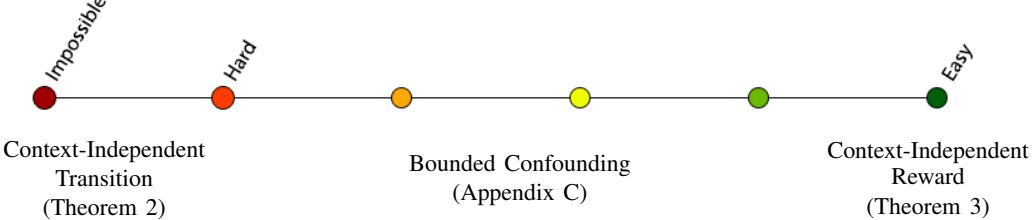

Figure 3: A spectrum for the difficulty of confounded imitation with covariate shift. Context-Independent transitions may result in non-identifiable and catastrophic candidates, whereas context-independent rewards reduce the problem to a standard imitation learning.

Recall that, in general, $\pi^* \in \Pi^*_{\mathcal{M}}$ may depend on the context $x \in \mathcal{X}$. Therefore, the set $\Upsilon_{\pi^*}$ corresponds to all deterministic policies that cannot be distinguished from $\pi^*$ based on the observed expert data. The following theorem shows that for any policy $\pi^* \in \Pi^*_{\mathcal{M}}$ and any policy $\pi_0 \in \Upsilon_{\pi^*}$, one could design a reward function $r_0$, for which $\pi_0$ is optimal, while the set $\Upsilon_{\pi^*}$ is indiscernible from $\Upsilon_{\pi_0}$, i.e., $\Upsilon_{\pi^*} = \Upsilon_{\pi_0}$ (see Appendix F for proof). In other words, $\Upsilon_{\pi^*}$ is the smallest set of candidate optimal policies.

**Theorem 1.** *[Sufficiency of $\Upsilon_{\pi^*}$] Assume $\rho_e \equiv \rho_o$. Let $\pi^* \in \Pi^*_{\mathcal{M}}$ and let $\pi_0 \in \Upsilon_{\pi^*}$. Then, $\Upsilon_{\pi^*} = \Upsilon_{\pi_0}$. Moreover, if $\pi_0 \neq \pi^*$, then there exists $r_0$ such that $\pi_0 \in \Pi^*_{\mathcal{M}_0}$ but $\pi^* \notin \Pi^*_{\mathcal{M}_0}$, where $\mathcal{M}_0 = (\mathcal{S}, \mathcal{A}, \mathcal{X}, P, r_0, \rho_o, \nu, \gamma)$.*

The above theorem shows that any policy in $\Upsilon_{\pi^*}$ is a candidate optimal policy, yet without knowing the context the expert used. Such ambiguity can result in selection of a suboptimal or even catastrophic policy. We provide a practical algorithm in Appendix B which calculates the ambiguity set $\Upsilon_{\pi^*}$, and returns an average policy, with computational guarantees. In the next subsection we analyze a more challenging scenario, for which $\rho_o \neq \rho_e$. In this case, $\Upsilon_{\pi^*}$ may not be sufficient for the imitation problem.

### 3.2 WITH COVARIATE SHIFT: $\rho_o \neq \rho_e$

Next, we assume covariate shift exists between the online environment and the expert data, i.e., $\rho_o \neq \rho_e$. Particularly, without further assumptions on the extent of covariate shift, we show two extremes of the problem. In Theorem 2 we prove that whenever the *transitions* are independent of the context, the data cannot in general be used for imitation. In contrast, in Theorem 3 we prove that, whenever the *reward* is independent of the context, the imitation problem can be efficiently solved.

Clearly, if $\text{Supp}(\rho_o) \nsubseteq \text{Supp}(\rho_e)$[1] then there exists $x \in \text{Supp}(\rho_o)$ for which $\pi^*$ is not identifiable from the expert data[2]. We therefore assume throughout that $\text{Supp}(\rho_o) \subseteq \text{Supp}(\rho_e)$. We begin by defining the set of non-identifiable policies as those that cannot be distinguished from their respective state-action frequencies without information on $\rho_e$.

**Definition 2.** *We say that $\{\pi_i\}_{i=1}^k$ are non-identifiable policies if there exist $\{\rho_i\}_{i=1}^k$ such that $d^{\pi_i}_{\rho_i}(s,a) = d^{\pi_j}_{\rho_j}(s,a)$ for all $i \neq j$.*

Next, focusing on catastrophic policies (recall Equation (1)), we define catastrophic expert policies as those which could be either optimal or catastrophic under $\rho_o$ for different reward functions.

**Definition 3.** *We say that $\{\pi_i\}_{i=1}^k$ are catastrophic expert policies if there exist $\{r_i\}_{i=1}^k$ such that for all $i$, $\pi_i \in \Pi^*_{\mathcal{M}_i}$, and $\exists j \in [k], j \neq i$ such that $\pi_i \in \Pi^{\dagger}_{\mathcal{M}_j}$, where $\mathcal{M}_j = (\mathcal{S}, \mathcal{X}, \mathcal{A}, P, r_j, \rho_o, \nu, \gamma)$.*

Using the fact that both $\rho_e$ and $r$ are unknown, the following theorem shows that whenever $P(s'|s,a,x)$ is independent of $x$, one could find two policies which are non-identifiable, catastrophic expert policies (see Appendix F for proof). In other words, in the case of context-independent transitions, without further information on $\rho_e$ or $r$ the expert data is useless for imitation. Furthermore, attempting to imitate the policy using the expert data could result in a catastrophic policy.

---

[1] For a distribution $\mathbb{P}$ we denote by $\text{Supp}(\mathbb{P})$ the support of $\mathbb{P}$.

[2] We define non-identifiability in Definition 2. We use a similar notion of identifiability as in Pearl (2009b)

**Theorem 2.** *[Catastrophic Imitation] Assume* $|\mathcal{X}| \geq |\mathcal{A}|$, *and* $P(s'|s, a, x) = P(s'|s, a, x')$ *for all* $x, x' \in \mathcal{X}$. *Then* $\exists \pi_{e,1}, \pi_{e,2}$ *s.t.* $\{\pi_{e,1}, \pi_{e,2}\}$ *are non-identifiable, catastrophic expert policies.*

While Theorem 2 shows the impossibility of imitation for context-free transitions, whenever the reward is independent of the context, the imitation problem becomes feasible. In fact, as we show in the following theorem, for context-free rewards, any policy in $\Upsilon_{\pi^*}$ is an optimal policy.

**Theorem 3.** *[Sufficiency of Context-Free Reward] Assume* $\text{Supp}(\rho_o) \subseteq \text{Supp}(\rho_e)$ *and* $r(s, a, x) = r(s, a, x')$ *for all* $x, x' \in \mathcal{X}$. *Then* $\Upsilon_{\pi^*} \subseteq \Pi^*_{\mathcal{M}}$.

Theorems 2 and 3 suggest that the hardness of the imitation problem under covariate shift lies on a wide spectrum (as depicted in Figure 3). While dependence of the transition $P(s'|s, a, x)$ on $x$ provides us with information to identify $x$ in the expert data, the dependence of the reward $r(s, a, x)$ on $x$ increases the degree of confounding in the imitation problem. Both of these results are concerned with arbitrary confounding.

**Bounded Confounding: A Sensitivity Perspective.** A common approach in causal inference is to bound the bias of unobserved confounding through sensitivity analysis (Hsu & Small, 2013; Namkoong et al., 2020; Kallus & Zhou, 2021). In our setting, this confounding bias occurs due to a covariate shift of the unobserved covariates. As we've shown in Theorem 2, though these covariates are observed in the online environment, their shifted and unobserved distribution in the offline data can render catastrophic results. Therefore, we consider the odds-ratio bounds of the sensitivity in distribution between the online environment and the expert data, as stated formally below.

**Assumption 1** (Bounded Sensitivity). *We assume that* $\text{Supp}(\rho_e) \subseteq \text{Supp}(\rho_o)$ *and that there exists some* $\Gamma \geq 1$ *such that for all* $x \in \text{Supp}(\rho_e)$, $\Gamma^{-1} \leq \frac{\rho_o(x)(1-\rho_e(x))}{\rho_e(x)(1-\rho_o(x))} \leq \Gamma$.

Next, we define the notion of $\delta$-ambiguity, a generalization of the ambiguity set in Definition 1.

**Definition 4** ($\delta$-Ambiguity Set). *For a policy* $\pi \in \Pi$, *we define the set of all deterministic policies that are* $\delta$-*close to* $\pi$ *by* $\Upsilon_\pi^\delta = \left\{ \pi' \in \Pi_{det} : \left| d_{\rho_o}^{\pi'}(s, a) - d_{\rho_e}^\pi(s, a) \right| < \delta, s \in \mathcal{S}, a \in \mathcal{A} \right\}$.

Similar to Definition 1, the $\delta$-ambiguity set considers all deterministic policies with a marginalized state-action frequency of distance at most $\delta$ from $\pi$. The following results shows that $\Upsilon_{\pi^*}^{\Gamma-1}$ is a sufficient set of candidate optimal policies, as long as Assumption 2 holds for some $\Gamma \geq 1$.

**Theorem 4.** *[Sufficiency of* $\Upsilon_{\pi^*}^{\Gamma-1}$*] Let Assumption 2 hold for some* $\Gamma \geq 1$. *Then* $\pi^* \in \Upsilon_{\pi^*}^{\Gamma-1}$.

For the interested reader, we further analyze the case of bounded confounding in Appendix C. We also demonstrate the effect of bounded confounding in Section 5. In the following section, we show that, while arbitrary confounding may result in catastrophic results for the imitation learning problem, when coupled with reward, one can still make use of the expert data.

## 4 USING EXPERT DATA WITH UNOBSERVED CONFOUNDERS FOR RL

In the previous section we showed sufficient conditions under which imitation is possible, with and without covariate shift. When covariate shift is present, but unknown, the imitation learning problem may be hard, or even impossible (see Theorem 2, catastrophic imitation). We ask, had we had access to the reward function, would the expert data be useful under arbitrary covariate shift? In this section we show that expert data with unobserved confounders can be used to converge to an expert policy, even when arbitary covariate shift is present. In our experiments (Section 5) we empirically show that using our method can also improve overall performance.

We view the confounded expert data as side information to the RL problem. Specifically, we assume access to the true reward signal in the online environment and wish to leverage the offline expert data to aid the agent in converging to an optimal policy. To do this, we define an optimization problem that maximizes the cumulative reward, while minimizing an $f$-divergence (e.g., KL-divergence, TV-distance, $\chi^2$-divergence) of state-action frequencies in $\Upsilon_{\pi^*}$,

$$\max_{\pi \in \Pi} \mathbb{E}_{x \sim \rho_o, s, a \sim d^\pi(s, a|x)}[r(s, a, x)] - \lambda D_f(d_{\rho_o}^\pi(s, a) || d_{\rho_e}^{\pi^*}(s, a)). \tag{P1}$$

**Algorithm 1** RL using Expert Data with Unobserved Confounders (Follow the Leader)

1: **input:** Expert data with missing context $\mathcal{D}^*$, $\lambda > 0$, policy optimization algorithm `ALG-RL`
2: **init:** Policy $\pi^0$
3: **for** $k = 1, \dots$ **do**
4: $\quad \rho_s \leftarrow \arg\min_\rho D_{KL}(d_{\rho_o}^{\pi^{k-1}}(s,a) || d_\rho^{\pi^*}(s,a))$
5: $\quad g^k \leftarrow \frac{1}{k}\left( g^{k-1} + \mathbb{E}_{s,a \sim d_{\rho_o}^{\pi^{k-1}}}\left[ \frac{1}{d_{\rho_s}^{\pi^*}(s,a)} \right] \right)$
6: $\quad \pi^k \leftarrow$ `ALG-RL`$(r(s,a,x) - \lambda g^k(s,a))$
7: **end for**

**Algorithm 2** RL using Expert Data with Unobserved Confounders (Online Gradient Descent)

1: **input:** Expert data with missing context, $\lambda, B, N > 0$, policy optimization alg. `ALG-RL`
2: **init:** Policy $\pi^0$, bonus reward network $g_\theta$
3: **for** $k = 1, \dots$ **do**
4: $\quad \rho_s \leftarrow \arg\min_\rho D_f(d_{\rho_o}^{\pi^{k-1}}(s,a) || d_\rho^{\pi^*}(s,a))$
5: $\quad$ **for** $e = 1, \dots N$ **do**
6: $\quad\quad$ Sample batch $\{s_i, a_i\}_{i=1}^B \sim d_{\rho_o}^{\pi^{k-1}}(s,a)$
7: $\quad\quad$ Sample batch $\{s_i^e, a_i^e\}_{i=1}^B \sim d_{\rho_s}^{\pi^*}(s,a)$
8: $\quad\quad$ Update $g_\theta$ according to $\nabla_\theta L(\theta) = \frac{1}{B}\sum_{i=1}^B \nabla_\theta[f^*(g_\theta(s_i^e, a_i^e)) - g_\theta(s_i, a_i)]$
9: $\quad$ **end for**
10: $\quad \pi^k \leftarrow$ `ALG-RL`$(r(s,a,x) - \lambda g_\theta(s,a))$
11: **end for**

Here, $\lambda > 0$ and $D_f$ is the $f$-divergence, where $f$ is a convex function $f : (0, \infty) \mapsto \mathbb{R}$. The solution to Problem (P1) is an optimal policy $\pi^* \in \Pi_{\mathcal{M}}^*$ as long as $\rho_o \equiv \rho_e$. Rewriting $D_f$ using its variational form (see Appendix A for background on the variational form of $f$-divergences), we get the following equivalent optimization problem, motivated by Nachum et al. (2019):

$$\max_{\pi \in \Pi} \min_{g: \mathcal{S} \times \mathcal{A} \mapsto \mathbb{R}} \mathbb{E}_{x \sim \rho_o, s, a \sim d^\pi(s,a|x)}[r(s,a,x) + \lambda g(s,a)] - \lambda \mathbb{E}_{s,a \sim d_{\rho_e}^{\pi^*}(s,a)}[f^*(g(s,a))], \quad \text{(P1b)}$$

where $f^*$ is the convex conjugate of $f$, i.e., $f^*(y) = \sup_x xy - f(y)$.

Unfortunately, when covariate shift exists (i.e., $\rho_o \neq \rho_e$), Problems (P1) and (P1b) are not ensured to converge to an optimal policy (Theorem 2). Instead, we propose to reformulate Problem (P1b) using a distribution $\rho_s$ which minimizes the $f$-divergence, as follows,

$$\max_{\substack{\pi \in \Pi \\ \rho_s \in \mathcal{B}(\mathcal{X})}} \min_{g: \mathcal{S} \times \mathcal{A} \mapsto \mathbb{R}} \mathbb{E}_{x \sim \rho_o, s, a \sim d^\pi(s,a|x)}[r(s,a,x) + \lambda g(s,a)] - \lambda \mathbb{E}_{s,a \sim d_{\rho_s}^{\pi^*}(s,a)}[f^*(g(s,a))]. \quad \text{(P2)}$$

Here, $\mathcal{B}(\mathcal{X})$ denotes the set of probability measures on the Borel sets of $\mathcal{X}$, and $d_{\rho_s}^{\pi^*}(s,a) = \mathbb{E}_{x \sim \rho_s}\left[ d^{\pi^*}(s,a \mid x) \right]$. Indeed, whenever $\text{Supp}(\rho_o) \subseteq \text{Supp}(\rho_e)$, we have that $(\pi, \rho_s) = (\pi^*, \rho_o)$ is a solution to Problem (P2). That is, unlike Problems (P1) and (P1b), Problem (P2) can achieve an optimal solution to the RL problem which still uses the expert data.

**Corrective Trajectory Sampling (CTS).** Solving Problem (P2) involves an expectation over an unknown distribution, $d_{\rho_s}^{\pi^*}(s,a)$. Fortunately, $d_{\rho_s}^{\pi^*}(s,a)$ can be equivalently written as an expectation over trajectories in $\mathcal{D}^*$, rather than expectation over unobserved contexts, as shown by the following proposition (see Appendix F for proof):

**Proposition 1.** *[Trajectory Sampling Equivalence] Let $\rho_s^*$ which minimizes Problem (P2) for some $\pi \in \Pi, g : \mathcal{S} \times \mathcal{A} \mapsto \mathbb{R}$, and assume $Supp(\rho_o) \subseteq Supp(\rho_e)$. Then, there exists $p^n \in \Delta_n$ such that $d_{\rho_s^*}^{\pi^*}(s,a) = \lim_{n \to \infty} \mathbb{E}_{i \sim p^n}\left[ (1-\gamma) \sum_{t=0}^\infty \gamma^t \mathbf{1}\{(s_t^i, a_t^i) = (s,a)\} \right]$.*

Proposition 1 allows us to estimate the inner minimization problem over $\rho_s$ in Problem (P2) using trajectory samples. Particularly, we uniformly sample $k$ distributions $p_1^n, \dots p_k^n$, where $p_j^n \in \Delta_n$, and then estimate

$$\min_{\rho_s} D_f(d_{\rho_o}^\pi || d_{\rho_s}^{\pi^*}) \approx \min_{j \in \{1, \dots, k\}} \left\{ D_f\left( d_{\rho_o}^\pi(s,a) \,\Big|\Big|\, \mathbb{E}_{i \sim p_j^n}\left[ \sum_{t=0}^\infty \gamma^t \mathbf{1}\{(s_t^i, a_t^i) = (s,a)\} \right] \right) \right\}, \quad (2)$$

which can be estimated by using the variational form of $D_f$ (see Appendix A). We call this procedure Corrective Trajectory Sampling (CTS), as it uses complete trajectory samples to account for the unknown context distribution $\rho_e$.

**Solving Problem (P2).** Algorithm 1 provides an iterative procedure for solving the optimization problem in Problem (P2). It uses alternative updates of a cost player (line 5) and policy player (line 6). In line 5 the gradient of $D_{KL}$ w.r.t. $d^\pi$ is taken using a Follow the Leader (FTL) cost player

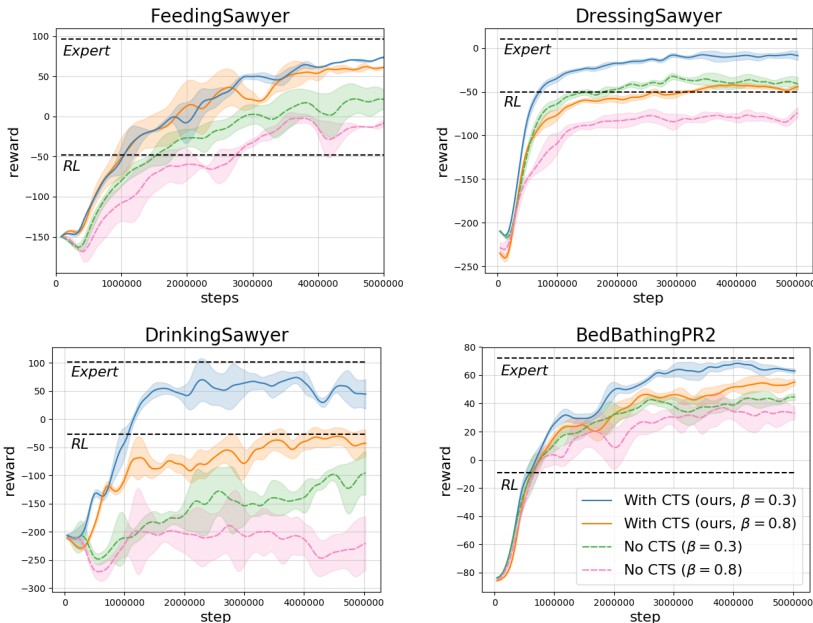

Figure 4: Plots compare training curves of using CTS vs. normal sampling of expert data for small ($\beta = 0.3$) and large ($\beta = 0.8$) covariate shift bias in four assistive-healthcare tasks. Dashed black lines show expert and RL (without data) scores. Runs were averaged over 5 seeds. Legend is shared across all plots.

to estimate the next bonus iterate. Finally, in line 6, an efficient, approximate policy optimization algorithm `ALG-RL` is executed using an augmented reward. The following theorem, provides convergence guarantees for Algorithm 1 (see Appendix F for proof based on Zahavy et al. (2021)).

**Theorem 5.** *Let* `ALG-RL` *be an approximate best response player that solves the RL problem in iteration $k$ to accuracy $\epsilon_k = \frac{1}{\sqrt{k}}$. Then, Algorithm 1 will converge to an $\epsilon$-optimal solution to Problem (P2) in $\mathcal{O}\left(\frac{1}{\epsilon^4}\right)$ samples.*

Notice that, while Theorem 5 shows Algorithm 1 converges to an optimal policy, it does not determine whether the expert data improves overall learning efficiency. We leave this theoretical question for future work. Nevertheless, in the following section we conduct extensive experiments to show that such data can indeed improve overall performance on various tasks.

A drawback of Algorithm 1 is that it needs to estimate the state-action frequencies. A practical implementation of Algorithm 1 using online gradient descent (OGD) is provided in Algorithm 2 – the algorithm does not require approximate estimates of the state-action frequencies, but rather, only the ability to sample from them. Similar to Algorithm 1, we use CTS (see Equation (2)) to estimate $\rho_s$ in line 4 according to some $f$-divergence. Here, samples are drawn from the current policy as well as samples from $\mathcal{D}^*$ (with CTS). We write $D_f$ in its variational form, and use a neural network representation for $g_\theta$. We then use the aforementioned samples to minimize the $f$-divergence using OGD. Finally, the policy is updated using `ALG-RL` and an augmented reward.

## 5 EXPERIMENTS

We tested our proposed approach for using expert data with hidden confounding in recommender-system and assistive-healthcare environments. For all our experiments we used $\chi^2$-divergence as our choice of $f$-divergence, as we found it to work best. Comparison to other divergences is provided in Figure 5 (left). We used PPO (Schulman et al., 2017) implemented in RLlib (Liang et al., 2018) for both the imitation as well as RL settings. We include specific choice of hyperparameters and an exhaustive overview of further implementation details in Appendix E.

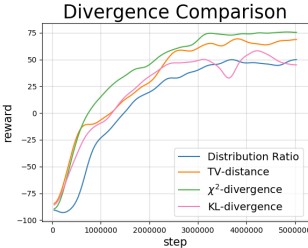 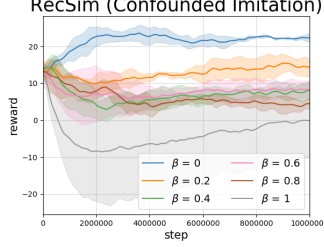 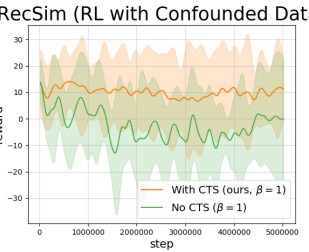

Figure 5: Left plot shows comparison of different choices of $f$-divergences for pure imitation (without reward and without covariate shift) on the BedBathingPR2 environment. Middle plot depicts execution of imitation with hidden confounding (without reward) for different levels of covariate shift. Right plot compares our CTS correction on the RecSim environment with strong covariate shift bias. All runs were averaged over 5 seeds.

**Assistive Healthcare.** Consider the challenge of providing physical assistance to disabled persons. A recently proposed set of tasks for assistive-healthcare, simulating autonomous robots as versatile caregivers (Erickson et al., 2020). Each task has a unique goal, affected by both the physical world as well as the patient specific preferences and disabilities.

We tested our algorithm on four tasks: feeding, dressing, bathing, and drinking. In these, we used the following features to define user context: gender, mass, radius, height, patient impairment, and patient preferences. The patient's mass, radius, and height distributions were dependent on gender. The patient's impairment was given by either limited movement, weakness, or tremor (with sporadic movement). Finally, the patient's preferences were affected by the velocity and pressure of touch forces applied by the robot. For the context distribution $\rho_o$ we used the default values as provided by the original environment. To enforce a distribution shift in the expert data, we shifted each distribution randomly with an additive factor $\beta \cdot \tilde{d}_x$, where $\beta \in [0, 1]$, and $\tilde{d}_x$ was a random distribution chosen from a set of shifting distributions (see Appendix E). Here $\beta$ corresponds to the covariate shift strength. The expert data was generated by a fixed policy trained using a dense reward function. A sparse reward signal was used for executing our experiments with the confounded expert data. For further details, we refer the reader to Appendix E.

Figure 4 depicts results for executing Algorithm 2 on four assistive-gym environments with various covariate shift strengths. As evident in most of the enviornments, covariate shift strongly affected overall performance. Particularly in the feeding, drinking, and dressing environments, the success of reaching the goal (i.e., spoon to mouth, cup to mouth, and sleeve to hand) was highly affected by the degree of covariate shift. This is due to the changing distribution of size, movement, and preferences of the patient, and thus of the goal. Nevertheless, in all environments, using the expert data (with and without CTS) was found to help induce better policies than executing the same RL algorithm without expert data. This suggests that expert data can assist in improving overall RL performance, yet correcting for covariate shift may significantly improve it in these domains.

**Recommender Systems.** In practical recommender systems, sequential interaction with users presents a great challenge for optimizing user long-term engagement and overall satisfaction (Ie et al., 2019). Leveraging expert data collected using, e.g., surveys to users, may greatly benefit future solutions. Because features are repeatedly added to these systems, full information in the data is rarely available. Here, we use the recently proposed RecSim Interest Evolution environment (Ie et al., 2019), simulating sequential user interaction with a slate-based recommender system. The environment consists of a document model for sampling documents, a user model for defining a distribution over user context features, and a user choice model, which defines the intent of the user based on observable document features and the user's sampled context (e.g., personality, satisfaction, interests, demographics, and other behavioral features such as session length or visit frequency).

We used a slate of 10 documents and a user context of dimension 20. To test the severity of the implications of Theorem 2 in the confounded imitation setting, we used a user-model sampled from a Beta-distribution. Specifically, for the expert data the user context features $x = (x_0, \ldots, x_{19})$ were sampled from a Beta-distribution, where $x_i \sim Beta(\alpha_i, 4)$, and $\alpha_i = 1.5 + \frac{8.5}{19}i$. In contrast, the online environment features were sampled from a shifted Beta-distribution with $\alpha_i = (1 - \beta)\left(1.5 + \frac{8.5}{19}i\right) + \beta\left(10 - \frac{8.5}{19}i\right)$, where $\beta \in [0, 1]$ defined the shift strength. While the

original environment used a uniform distribution to generate user contexts, the Beta-distribution let us analyze severe forms of covariate shifts, testing the limits of our results in Sections 3 and 4.

Figure 5(a) depicts the effect of increased covariate shift on imitation in the RecSim environment with a dataset of 100 expert trajectories (generated by an optimal policy that had access to the full context). Without covariate shift ($\beta = 0$) an optimal score is achieved, and as $\beta$ increases, performance decreases. Particularly, as the mirrored distribution is reached ($\beta = 1$), a catastrophic policy is reached. While the imitator "believes" to have reached an optimal policy, it has in fact reached a catastrophic one, as shown by the orange plot. Conversely, Figure 5(b) depicts the benefit of using confounded expert data in the RL setting (with an online reward signal). Though strong confounding is present, the agent is capable of leveraging the data to improve overall learning performance.

## 6   RELATED WORK

**Imitation Learning.** The imitation learning problem has been extensively studied in both the fully offline (Pomerleau, 1989; Bratko et al., 1995) as well as online setting (Ho & Ermon, 2016; Fu et al., 2018; Kim & Park, 2018; Brantley et al., 2019). Specific to our work are GAIL (Ho & Ermon, 2016), AIRL (Fu et al., 2017), and DICE (Kostrikov et al., 2019), which use distribution matching methods. Our work generalizes these settings to imitation with hidden confounders.

**Reinforcement Learning with Expert Data.** Much work has revolved on leveraging offline data for RL. Recently, offline RL (Levine et al., 2020) has shown great improvement over regular offline imitation techniques (Kumar et al., 2020; Kostrikov et al., 2021; Tennenholtz et al., 2021a; Fujimoto & Gu, 2021). In the online RL setting, the combination of offline data to improve RL efficiency has shown great success (Nair et al., 2020). KL-regularized techniques (Peng et al., 2019; Siegel et al., 2019) as well as DICE-based algorithms (Nachum et al., 2019) have also shown efficient utilization of offline data. Our work generalizes the latter to the confounded setting.

**Intersection of Causal Inference and Imitation Learning.** Closely related to our work is that of Zhang et al. (2020). There, the authors suggest a notion of imitability, showing when observational data can help identify a policy under some partially observed structural causal model. Our work provides an alternative perspective on the problem. In contrast to their work, we rely on concurrent imitation approaches (i.e. state-action frequency matching techniques) and importantly, allow access to the online environment. Furthermore, we provide guarantees and practical algorithms for both the imitation and RL settings. We refer the reader to Appendix D for an interpretation of our framework in CI terminology, from a perspective of stochastic interventions in a Structural Causal Model.

Another intersection with causal inference discusses the problem of causal confusion in imitation (de Haan et al., 2019). Causal confusion is concerned with the problem of nuisances in *observed* confounded data due to an unknown causal structure. These "causal misidentifications" can lead to spurious correlations and catastrophic failures in generalization. In contrast, our work discusses the orthogonal problem of hidden confounders with possible covariate shift.

**Intersection of Causal Inference and Reinforcement Learning.** Previous work has analyzed the problem of optimal control from logged data with unobserved confounders (Lattimore et al., 2016), as well as utilizing (non-expert) confounded data for online interactions (Tennenholtz et al., 2021b). Much work has revolved around the reinforcement learning setup with access to (non-expert) confounded data (Zhang & Bareinboim, 2019; Wang et al., 2020). Other work has considered the problem of off-policy evaluation from confounded data (Tennenholtz et al., 2020; Oberst & Sontag, 2019; Kallus & Zhou, 2020). Our work is focused on leveraging *expert* data with hidden confounders and possible covariate shift in both the imitation and the RL settings.

## 7   CONCLUSION

This work presented and analyzed the problem of using expert data with hidden confounders for both the imitation and RL settings. We showed that covariate shift of hidden confounders between the expert data and the online environment can result in learning catastrophic policies, rendering imitation learning hard, or even impossible (Theorem 2). In addition, we showed that when a reward is provided, using the expert data is still possible under arbitrary hidden covariate shift (Theorem 5), and proposed new algorithms for tackling this problem using corrective trajectory sampling (CTS).

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

| Distribution Matching | Equivalent Representation | Comments |
|---|---|---|
| Distribution Ratio | $\sup_{g:\mathcal{S}\times\mathcal{A}\mapsto(0,1)} \mathbb{E}_{s,a\sim d_{\rho_e}^{\pi'}}[\log(g(s,a))] + \mathbb{E}_{s,a\sim d_{\rho_o}^{\pi}}[\log(1-g(s,a))]$ | GAIL (Ho & Ermon, 2016) |
| KL-divergence | $\sup_{g:\mathcal{S}\times\mathcal{A}\mapsto\mathbb{R}} \mathbb{E}_{s,a\sim d_{\rho_o}^{\pi}}[g(s,a)] - \log\mathbb{E}_{s,a\sim d_{\rho_e}^{\pi'}}\left[e^{g(s,a)}\right]$ | Donsker-Varadhan Representation (Kostrikov et al., 2019) |
| $\chi^2$-divergence | $\sup_{g:\mathcal{S}\times\mathcal{A}\mapsto\mathbb{R}} 2\mathbb{E}_{s,a\sim d_{\rho_o}^{\pi}}[g(s,a)] - \mathbb{E}_{s,a\sim d_{\rho_e}^{\pi'}}\left[g^2(s,a)\right]$ | Variational Representation of $f$-Divergence |
| TV Distance | $\sup_{|g|\leq\frac{1}{2}} \mathbb{E}_{s,a\sim d_{\rho_o}^{\pi}}[g(s,a)] - \mathbb{E}_{s,a\sim d_{\rho_e}^{\pi'}}[g(s,a)]$ | Variational Representation of $f$-Divergence |

Table 1: Different distribution matching techniques and their equivalent representations.

## APPENDIX

## A  BACKGROUND: DISTRIBUTION MATCHING FOR IMITATION LEARNING

A common approach used in (non-confounded) imitation learning is matching the policy's state-action frequency $d_{\rho_o}^{\pi}$ to the offline target distribution $d_{\rho_e}^{\pi^*}$. Consider a source distribution $p \in \Delta_N$ and target distribution $q \in \Delta_N$. GAIL (Ho & Ermon, 2016) uses the distribution ratio objective $log(p/q)$, which can estimated using a GAN-like objective $D_R(p||q) = \sup_{g:\mathcal{Z}\mapsto(0,1)} \mathbb{E}_p[\log(g(z))] + \mathbb{E}_q[\log(1-g(z))]$, to match the distribution $p$ to $q$.

This technique can be generalized to $f$-divergences (Csiszár & Shields, 2004; Liese & Vajda, 2006; Kostrikov et al., 2019; Ke et al., 2020). Specifically, we wish to minimize a discrepancy measure from $p$ to $q$, namely $\min_{p\in\mathcal{K}} D(p||q)$. For a convex function $f : [0:\infty) \mapsto \mathbb{R}$, the $f$-divergence of $p$ from $q$ is defined by $D_f(p||q) = \mathbb{E}_q\left[f\left(\frac{p}{q}\right)\right]$. DICE (Kostrikov et al., 2019) uses the variational representation of the $f$-divergence,

$$D_f(p||q) = \sup_{g:\mathcal{Z}\mapsto\mathbb{R}} \mathbb{E}_p[g(z)] - \mathbb{E}_q[f^*(g(z))],$$

where $f^*$ is the Fenchel conjugate of $f$ defined by $f^*(y) = \sup_x xy - f(y)$. The convex conjugate has closed form solutions for the total variation distance, KL-divergence, $\chi^2$-divergence, Squared Hellinger distance, Le Cam distance, and Jensen-Shannon divergence. Using the variational representation of the $f$-divergence we can estimate $D_f$ using samples from $p$ and $q$. Table 1 presents examples of various $f$-divergences and their respective dual formulation. We also add the distribution ratio for comparison to the table, though it is not an $f$-divergence.

## B  CONFOUNDED IMITATION - ALGORITHM AND CONVERGENCE GUARANTEES

### B.1  A TOY EXAMPLE

To gain intuition, we start with a simple toy example. Consider the three-state example depicted in Figure 6. Here, the environment initiates at state $A$ w.p. 1, after which the agent can choose to (deterministicaly) transition to state $B$ or $C$. The agent then receives a reward depending on the context. The optimal policy is given by $\pi^*(a|s,x) = \mathbf{1}\{a = a_B, x = x_1\} + \mathbf{1}\{a = a_C, x = x_2\}$ for $s = A$, and any action is optimal for $s \neq A$. Without loss of generality we assume $\pi^*(a_B|B,x) = \pi^*(a_C|C,x) = 1$. We turn to analyze the marginalized state-action frequency, which uniquely defines the set of optimal policies (Puterman, 2014). Denoting $\rho_e(x_1) = \rho$, we have that $d_{\rho_e}^{\pi^*}(s,a) = \rho d^{\pi^*}(s,a|x_1) + (1-\rho)d^{\pi^*}(s,a|x_2)$. Then, $d_{\rho_e}^{\pi^*}(s,a) = (1-\gamma)\mathbf{1}\{s = A\} + \rho\gamma\mathbf{1}\{s = B, a = a_B\} + (1-\rho)\gamma\mathbf{1}\{s = C, a = a_C\}$.

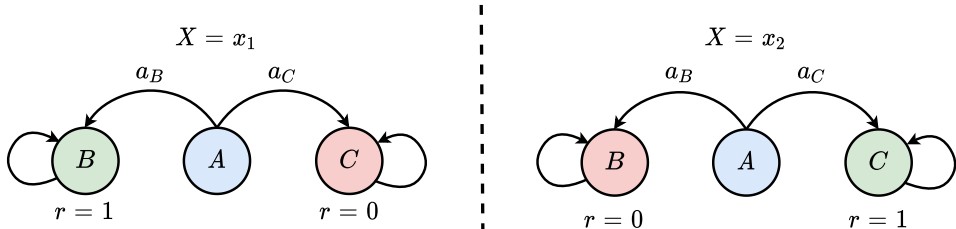

Figure 6: A contextual MDP with state space $\mathcal{S} = \{A, B, C\}$, action space $\mathcal{A} = \{a_B, a_C\}$ and context space $\mathcal{X} = \{x_1, x_2\}$. We assume $\nu(A|x) = 1$ for all $x \in \mathcal{X}$. The actions $a_B, a_C$ transition the agent to states $B, C$, respectively, after which the agent receives a reward $r \in \{0, 1\}$ depending on the context. We assume $B, C$ are sink states.

---

**Algorithm 3** Confounded Imitation

1: **input:** Expert data with missing context $\mathcal{D}^* \sim d_{\rho_e}^{\pi^*}$, $\lambda > 0$, sensitivity bound $\delta \geq 0$.
2: **init:** $\Upsilon = \emptyset$
3: **for** $n = 1, \dots$ **do**
4:    Sample $u(s, a) \sim U[0, \delta], \forall s, a$
5:    $L^*(\pi; g_0) := \mathbb{E}_{s,a \sim d_{\rho_o}^{\pi}(s,a)}[g_0(s, a)] - \mathbb{E}_{s,a \sim d_{\rho_e}^{\pi^*}(s,a) + u(s,a)}[g_0(s, a)]$
6:    $L_i(\pi; g_i) := \mathbb{E}_{x \sim \rho_o, s, a \sim d^{\pi}(s,a|x)}[g_i(s, a, x)] - \mathbb{E}_{x \sim \rho_o, s, a \sim d^{\pi_i}(s,a|x)}[g_i(s, a, x)] \quad , i \geq 1$
7:    Compute $\pi_n$ by solving

$$\min_{\pi \in \Pi_{\text{det}}} \max_{|g_0| \leq \frac{1}{2}, |g_i| \leq \frac{1}{2}} \left\{ L^*(\pi; g_0(s, a)) - \lambda \min_i L_i(\pi; g_i(s, a, x)) \right\} \tag{4}$$

8:    **if** $\pi_n \in \Upsilon$ **then**
9:       Terminate and return $\bar{\pi}(a|s, x) = \frac{\sum_{i=1}^{n-1} d^{\pi_i}(s,a,x)}{\sum_{i=1}^{n-1} \sum_{a'} d^{\pi_i}(s,a',x)}$
10:    **else**
11:       $\Upsilon = \Upsilon \cup \{\pi_n\}$
12:    **end if**
13: **end for**

---

**No Covariate Shift.** Suppose $\rho_o = \rho_e$, and $\rho = \frac{1}{2}$. Trivially $d_{\rho_e}^{\pi^*}(s, a) = d_{\rho_o}^{\pi^*}(s, a)$. We define the (suboptimal) policy

$$\pi_0(a|A, x) = 1 - \pi^*(a|A, x) \quad , a \in \mathcal{A}, x \in \mathcal{X}. \tag{3}$$

It can be verified that $d_{\rho_e}^{\pi^*}(s, a) = d_{\rho_o}^{\pi_0}(s, a)$ still holds, yet $\pi_0$ is catastrophic (Equation (1)) with value zero. A question arises: can we show that $\pi_0$ is a suboptimal policy given access to the expert data (i.e., access to $d_{\rho_e}^{\pi^*}(s, a)$) and a forward model $P(s'|s, a, x)$?

Unfortunately, one cannot prove that $\pi_0$ is suboptimal. Informally, notice that $\pi_0$ is an optimal policy for an alternative reward function, $r_0(s, a, x) = 1 - r(s, a, x)$, yet is catastrophic w.r.t. the true reward $r$. Indeed, since $r$ is unknown and $d_{\rho_o}^{\pi_0}(s, a) = d_{\rho_o}^{\pi^*}(s, a)$, we cannot reject $r_0$ (i.e., we cannot conclude that $r_0$ is not the true reward). In other words, one cannot use the data to differentiate which of $\{\pi_0, \pi^*\}$ is the optimal policy.

**With Covariate Shift.** Next, assume $\rho_o \neq \rho_e$, and define $\pi_0$ as in Equation (3). Let $\widetilde{\rho_e} = 1 - \rho_e$ and recall that $\rho_e(x_1) = \rho$. Then, we have that

$$d_{\widetilde{\rho_e}}^{\pi_0}(s, a) = (1 - \rho)d^{\pi_0}(s, a|x_1) + \rho d^{\pi_0}(s, a|x_2) = (1 - \rho)d^{\pi^*}(s, a|x_2) + \rho d^{\pi^*}(s, a|x_1) = d_{\rho_e}^{\pi^*}(s, a).$$

Indeed, the expert data is incapable of distinguishing $\pi_0$ and $\pi^*$, since $d_{\widetilde{\rho_e}}^{\pi_0} = d_{\rho_e}^{\pi^*}$, and $\rho_e$ is unknown. Unfortunately, as we've shown previously, $\pi_0$ achieves value zero. Notice that, unlike the previous section, one cannot distinguish $\pi^*$ from the catastrophic policy $\pi_0$ for *any choice* of $\rho_o$.

## B.2 A PRACTICAL ALGORITHM

The ambiguity set of Theorem 1 may contain suboptimal policies. Instead of randomly selecting a policy from this set, we can choose the average policy. The following proposition shows that such a selection is favorable.

**Proposition 2.** *Define the mean policy* $\bar{\pi}(a|s,x) = \frac{\sum_{\pi \in \Upsilon_{\pi^*}} d^{\pi}(s,a,x)}{\sum_{\pi \in \Upsilon_{\pi^*}} \sum_{a'} d^{\pi}(s,a',x)}$, *and denote* $\alpha^* = \frac{|\Pi^*_{\mathcal{M}} \cap \Upsilon_{\pi^*}|}{|\Upsilon_{\pi^*}|} \in [0,1]$. *Then,* $v_{\mathcal{M}}(\bar{\pi}) \geq \alpha^* v^* + (1 - \alpha^*) \min_{\pi \in \Upsilon_{\pi^*}} v_{\mathcal{M}}(\pi)$.

**Remark 1.** *Note that* $\bar{\pi}$ *is generally not the average policy* $\frac{1}{|\Upsilon_{\pi^*}|} \sum_{\pi \in \Upsilon_{\pi^*}} \pi(a|s,x)$.

**Remark 2.** *In an episodic setting,* $\bar{\pi}$ *can be estimated by uniformly sampling a policy* $\pi \in \Upsilon_{\pi^*}$ *at the beginning of the episode, and playing it until the environment terminates.*

Algorithm 3 describes our method for calculating the ambiguity set of Theorem 1, and returns $\bar{\pi}$ of Proposition 2. At every iteration of the algorithm, we find a new policy in the set by minimizing the total variation distance (written in variational form) between $d^{\pi^*}_{\rho_o}(s,a)$ and $d^{\pi}_{\rho_o}(s,a)$, while regularizing it with the distance between $\pi$ and all previously collected $\pi_i \in \Upsilon$. Algorithm 3 also uses a sensitivity parameter $\delta \geq 0$ (defined formally in Appendix C) whenever bounded covariate shift is present. For this section we assume $\delta = 0$.

In practice, the functions $L^*$ and $L_i$ in lines 4 and 5 are estimated using samples from trajectories of $\pi, \pi_i$, and $\mathcal{D}^*$. We then solve the min-max problem of Equation (4) using a parametric representations of $g_i$ and online gradient decent. The following proposition states that Algorithm 3 indeed retrieves the set $\Upsilon_{\pi^*}$.

**Proposition 3.** *Assume* $\rho_e \equiv \rho_o$ *and* $|\Upsilon_{\pi^*}| < \infty$. *Then there exists* $\lambda^* > 0$ *such that for any* $\lambda \in (0, \lambda^*)$, *Algorithm 3 (with* $\delta = 0$ *sensitivity) will return* $\bar{\pi}$ *of Proposition 2 after exactly* $|\Upsilon_{\pi^*}|$ *iterations.*

## B.3 IMITATION WITH CONTEXT-FREE REWARD

We tested Algorithm 3 on both the RecSim environment as well as a four-rooms environment with random instantiations of walls. Experiments for the RecSim environment are readily provided in Section 5. Here we describe our simple four-rooms environment and show experiments w.r.t. Theorem 3.

The four-rooms environment, as depicted in Figure 7, is a $15 \times 15$ grid-world in which an agent can take one of four actions: LEFT, RIGHT, UP, or DOWN. Each action moves the agent in the specified direction whenever no obstacle is present. The agent (shown in blue) must reach the (green) goal while avoiding the (red) mine. When the goal is reached the agent receives a reward of $+1$ and the episode terminates. In contrast, if the agent reaches the mine, she receives a reward of $-1$ and the episode terminates. The state space of the environment consists of the agent's (row, col) position in the world. The rest of the information in the environment is defined by the context $x$. Particularly, the context is defined by the position of the green goal, the position of the red mine, and the specific instantiation of walls (two instantiations are depicted in Figure 7).

We trained an agent with full information (i.e., observed context, including goal location, mine location, and walls). We generated expert data w.r.t. the trained agent. To demonstrate the result of Theorem 3 we executed Algorithm 3 with both a shifted distribution and the default distribution of walls. We did not change the distribution of goal and mine. Note that since the distribution of walls only affects the transition function and not the reward, we expect, by Theorem 3, the optimal solution to remain the same. Indeed, as shown in Figure 7 after training an agent with no access to the contextual information of the walls in the expert data, the agent achieved comparable results both with and without covariate shift on the distribution of walls.

This result seem surprising at first, as the walls are essential for solving the task at hand. Nevertheless, since the distribution of wall *is observed* in the online environment, the partially observed expert data suffices to obtain an optimal policy. This settles with Theorem 3 which indeed states that this information is not needed in the expert data in order to obtain an optimal policy.

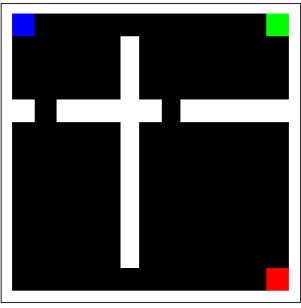
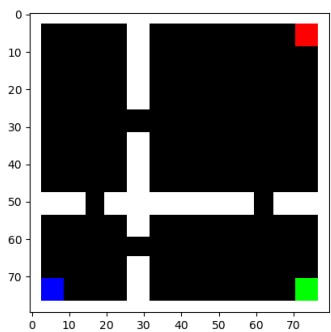

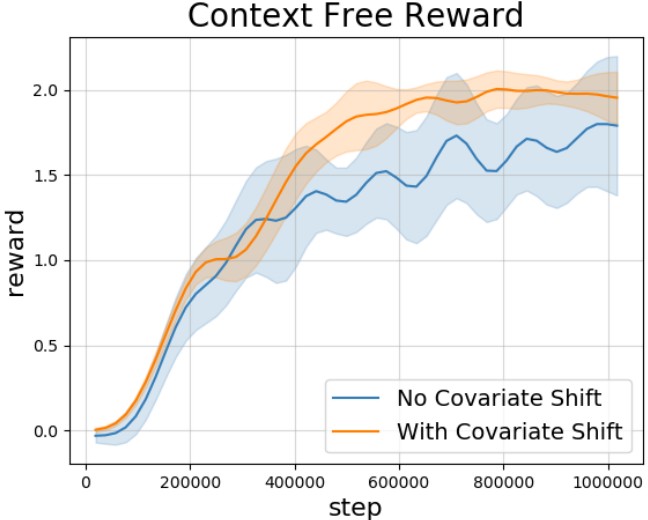

Figure 7: Results for the rooms environment with covariate shift affecting only the distribution of walls. It is evident that whenever the reward is context-free comparable performance is obtained. Runs averaged over 5 seeds.

## C  BOUNDED HIDDEN CONFOUNDING

In this section we discuss the imitation learning problem under bounded hidden confounders. There are several ways to define boundness of unobserved confounders. In Section 3 we showed that, under *arbitrary* covariate shift and context-free transitions, the imitation learning problem is impossible, i.e., one cannot rule out a catastrophic policy. We begin by considering the effect of bounded covariate shift, i.e., $\frac{\rho_o}{\rho_e} \leq C$. We then consider almost-context-free rewards, showing a tradeoff w.r.t. the hardness of the imitation problem.

**A Sensitivity Perspective.**  A common approach in causal inference is to bound the bias of unobserved confounding through sensitivity analysis (Hsu & Small, 2013; Namkoong et al., 2020; Kallus & Zhou, 2021). In our setting, this confounding bias occurs due to a covariate shift of the unobserved covariates. As we've shown in Theorem 2, though these covariates are observed in the online environment, their shifted and unobserved distribution in the offline data can render catastrophic results. Therefore, we consider the odds-ratio bounds of the sensitivity in distribution between the online environment and the expert data, as stated formally below.

**Assumption 2** (Bounded Sensitivity). *We assume that $Supp(\rho_e) \subseteq Supp(\rho_o)$ and that there exists some $\Gamma \geq 1$ such that for all $x \in Supp(\rho_e)$*

$$\Gamma^{-1} \leq \frac{\rho_o(x)(1 - \rho_e(x))}{\rho_e(x)(1 - \rho_o(x))} \leq \Gamma.$$

Next, we define the notion of $\delta$-ambiguity, a generalization of the ambiguity set in Definition 1.

**Definition 5** ($\delta$-Ambiguity Set). *For a policy $\pi \in \Pi$, we define the set of all deterministic policies that are $\delta$-close to $\pi$ by*

$$\Upsilon_\pi^\delta = \left\{ \pi' \in \Pi_{det} : \left| d_{\rho_o}^{\pi'}(s,a) - d_{\rho_e}^\pi(s,a) \right| < \delta, s \in \mathcal{S}, a \in \mathcal{A} \right\}.$$

Similar to Definition 1, the $\delta$-ambiguity set considers all deterministic policies with a marginalized state-action frequency of distance at most $\delta$ from $\pi$. The following results shows that $\Upsilon_{\pi^*}^{\Gamma-1}$ is a sufficient set of candidate optimal policies, as long as Assumption 2 holds for some $\Gamma \geq 1$.

**Theorem 6.** *[Sufficiency of $\Upsilon_{\pi^*}^{\Gamma-1}$] Let Assumption 2 hold for some $\Gamma \geq 1$. Then $\pi^* \in \Upsilon_{\pi^*}^{\Gamma-1}$.*

The above result suggests that Algorithm 3 can be executed over $\Upsilon_{\pi^*}^{\Gamma-1}$ by adding $\delta = \Gamma - 1$ additive uniform noise to $d_{\rho_e}^{\pi^*}(s,a)$ (see Line 4 of Algorithm 3), and executing the algorithm for a finite number of iterations, finally selecting an average policy from the approximate set.

**Context Reconstruction.**  When bounded covariate shift is present, one might attempt to learn an inverse mapping of contexts from observed trajectories in the data.

We denote by $P_\rho^\pi$ the probability measure over contexts $x \in \mathcal{X}$ and trajectories $\tau = (s_0, a_0, s_1, a_1, \ldots s_H)$ as induced by the policy $\pi$ and context distribution $\rho$. That is,

$$P_\rho^\pi(x, \tau) = \rho(x)\nu(s_0|x) \prod_{t=0}^{H-1} P(s_{t+1}|s_t, a_t, x)\pi(a_t|s_t, x).$$

As the true context is observed in the online environment, we can calculate for any $\pi$ the quantity $P_{\rho_o}^\pi(x, \tau)$. As the expert data distribution was generated by the marginalized distribution $P_{\rho_o}^{\pi^*}(\tau) = \sum_{x \in \mathcal{X}} P_{\rho_o}^{\pi^*}(x, \tau)$, it is unclear if knowledge of $P_{\rho_o}^\pi(x, \tau)$ is beneficial.

Fortunately, whenever Assumption 2 holds, a high probability of reconstructing a context in the online environment induces a high probability of reconstructing it in the expert data. To see this, assume that there exists $\delta \in [0, 1]$ such that for all $\pi \in \Upsilon_{\pi^*}^{\Gamma-1}$, $\tau \in \text{Supp}(P_{\rho_e}^\pi(\tau))$, there exists $x \in \mathcal{X}$ such that

$$P_{\rho_o}^\pi(x|\tau) \geq \min\{(1-\delta)(\rho_o(x) + \Gamma(1 - \rho_o(x))), 1\}. \tag{5}$$

That is, we assume that for any policy that $\delta$-ambiguous to $\pi^*$, and any induced trajectory of $x \in \mathcal{X}$, one can with high probability identify $x$ in the online environment. Importantly, this property can be verified in the online environment. When Assumption 2 and 5 hold, we get that

$$P_{\rho_e}^\pi(x|\tau) = \frac{P_{\rho_e}^\pi(\tau|x)\rho_e(x)}{P_{\rho_e}^\pi(\tau)} \geq \frac{P_{\rho_e}^\pi(\tau|x)}{P_{\rho_e}^\pi(\tau)} \frac{\rho_o(x)}{\rho_o(x) + \Gamma(1 - \rho_o(x))} = \frac{P_{\rho_o}^\pi(x|\tau)}{\rho_o(x) + \Gamma(1 - \rho_o(x))} \geq 1 - \delta.$$

In other words, we can reconstruct $x$ with probability $1 - \delta$ for any trajectory $\tau$ which satisfies the above. This allows us to deconfound essential parts of the expert data, rendering it useful for the imitation problem, even when reward is not provided. We leave further analysis of this direction for future work.

**Context-Dependent Reward.**  In Theorem 3 we showed that whenever the reward is independent of the context then the imitation problem is easy, in the sense that any policy $\pi_0 \in \Upsilon_{\pi^*}$ is also an optimal policy. Here, we relax the assumption on the reward, and instead assume bounded dependence of the reward on the context. The following definition upper bounds the confounding effect of the reward w.r.t. the context.

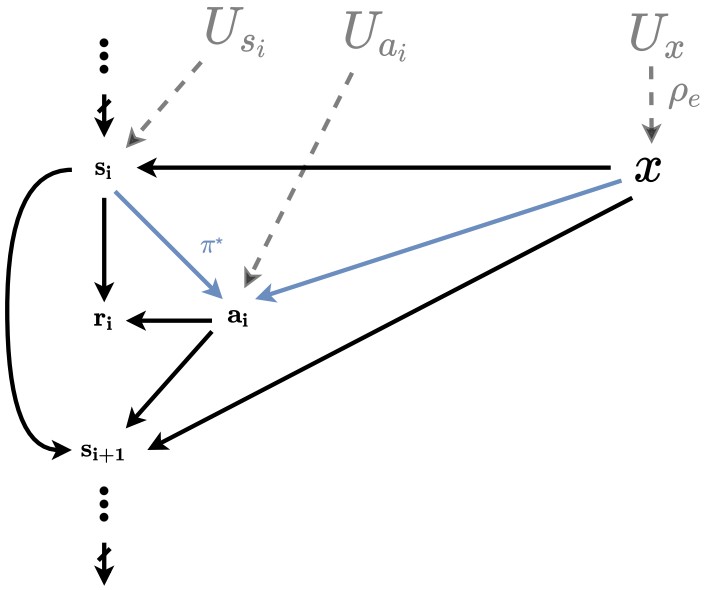

Figure 8: **Contextual MDP Causal Diagram**.

**Definition 6.** *Let* $\epsilon : \mathcal{X} \mapsto \mathbb{R}$ *such that*

$$\min_{r_0 : \mathcal{S} \times \mathcal{A} \mapsto \mathbb{R}} |r(s, a, x) - r_0(s, a)| \leq \epsilon(x) \quad , s \in \mathcal{S}, a \in \mathcal{A}, x \in \mathcal{X}$$

Using the above definition, we can now show that any policy in $\Upsilon_{\pi^*}$ is still approximately optimal, as shown by the following result.

**Theorem 7.** *[Context Dependent Reward]* *Let* $\epsilon : \mathcal{X} \mapsto \mathbb{R}$ *of Definition 6. Denote* $\epsilon_{oe} = \mathbb{E}_{x \sim \rho_o(x)}[\epsilon(x)] + \mathbb{E}_{x \sim \rho_e(x)}[\epsilon(x)]$. *Then for any* $\pi^* \in \Pi_{\mathcal{M}}^*$, $\pi_0 \in \Upsilon_{\pi^*}$

$$v(\pi_0) \geq v(\pi^*) - \epsilon_{oe}.$$

A direct corollary for the above result states that for $\epsilon : \mathcal{X} \mapsto \mathbb{R}$ of Definition 6, if $\epsilon(x) = \epsilon$, for all $x \in \mathcal{X}$, then for $\pi_0, \pi^*$ of Theorem 7, it holds that $v(\pi_0) \geq v(\pi^*) - 2\epsilon$. That is, $\pi_0$ is an approximately optimal policy.

## D    RELATION TO CAUSAL INFERENCE

Our work is focused on the problem of hidden confounders in expert data for imitation and reinforcement learning. We have chosen to write the paper in terminology famililar to the RL community. In this section we address and formalize the problem in Causal Inference (CI) terminology. We begin by defining Structural Causal Models (SCM, Pearl (2009a)) – a basic building block of our framework. We then show how the confounded imitation problem can be formalized as an intervention over a specific SCM. Generally speaking, the causal view casts the environment, namely the expert environment generating the offline data, and the online environment, as confounders.

**Definition 7** (Structural Causal Models). *A Structural Causal Model (SCM) is a tuple* $\mathcal{M} = (U, V, \mathcal{F}, P(U))$ *where* $U$ *is a set of exogenous variables and* $V$ *is a set of endogenous variables.* $\mathcal{F}$ *is a set of functions such that* $f_i \in \mathcal{F}$ *are functions mapping a set of endogenous variables* $Pa_i \subseteq V \backslash \{V_i\}$ *and a set of exogenous variables* $U_i \subseteq U$ *to the domain of* $V_i$, *i.e.,* $V_i = f_i(Pa_i, U_i)$. *Finally,* $P(u)$ *is a probability distribution over the set of exogenous variables* $U$. *We assume that the SCM is recursive, i.e., that the causal diagram associated with it is acyclic.*

Every SCM $\mathcal{M}$ is associated with a causal diagram $\mathcal{G}$, as depicted in Figure 8. Our framework relies largely on the formulation of stochastic interventions, as proposed in Correa & Bareinboim (2020).

We consider stochastic, conditional (non-atomic) interventions, defined by regime indicators $\sigma_Z$ (Pearl, 2000; Correa & Bareinboim, 2020), defined formally below.

**Definition 8** (Non-Atomic Interventions). *Given a SCM $\mathcal{M} = (U, V, \mathcal{F}, P(U))$ and a subset $Z \subseteq V$, an intervention $\sigma_Z = \{\sigma_{Z_1}, \ldots, \sigma_{Z_n}\}$ defines a new SCM $\mathcal{M}_{\sigma_Z} = (U, V, \mathcal{F}^*, P(U))$ in which the set of functions $\mathcal{F}$ is changed to $\mathcal{F}^* = \{f_i^*\}_{i:V_i \in \{Z_j\}_{j=1}^n} \bigcup \{f_i\}_{i:V_i \in V \setminus \{Z_j\}_{j=1}^n}$.*

Non-atomic interventions are a generalization of the classic atomic **do**$(X = x)$ interventions, defined by the SCM $\mathcal{M}_z$ and causal diagram $\mathcal{G}_{\overline{Z}}$ in which all edges incoming into $Z$ are removed. We have that

$$P(y|\mathbf{do}(Z = z)) = P(y|z; \sigma_Z = \mathbf{do}(Z = z)).$$

Atomic interventions replace function in $\mathcal{F}$ by constant functions, whereas non-atomic interventions use general functions. For notational simplicity, when a single intervention is applied to some $f_i \in \mathcal{F}$, we denote it by $\sigma_Z = \mathbf{d}(f_i \leftarrow f_i^*)$, indicating that in the interventional distribution $f_i^*$ is used instead of $f_i$. Next we define the identifiability of a causal effect under an intervention, as follows.

**Definition 9** (Identifiability). *Let $X, Y, Z \subseteq V$ with $Y \cap Z = \emptyset$ in some SCM with causal diagram $\mathcal{G}$. Given an intervention $\sigma_Z = \{\sigma_{Z_1}, \ldots, \sigma_{Z_n}\}$, the causal effect $P(y|x, \sigma_Z)$ is said to be identifiable from $V' \subseteq V$ if it can be uniquely computed from $P(V')$ for every assignment $(y, x)$ in every model that induces $\mathcal{G}$ and $P(V')$.*

Following the model definitions of Section 2, we define the contextual MDP SCM as follows.

**Definition 10.** *A contextual MDP SCM is defined by the causal diagram of Figure 8. For some horizon $H > 0$, the SCM is defined by the set of endogenous variables $V = \{s_i\}_{i=0}^H \cup \{a_i\}_{i=0}^H \cup \{x\} \cup \{r_i\}_{i=0}^H$ (denoting the states, actions, context and rewards, respectively), a set of exogenous variables $U$, and functions $\mathcal{F} = \{f_{s_i}, f_{a_i}, f_{r_i}, f_{\rho_e}, f_{\nu_0}\}$, where $f_{s_i}$ correspond to the transition function, $f_{a_i}$ the expert policy, $f_{r_i}$ the reward function, $f_{\rho_e}$ the context expert distribution, and $f_{\nu_0}$ the initial context-dependent state distribution.*

Relating to our formal definition of our model in Section 2, with slight abuse of notations, the functions $\{f_s, f_g, f_a, f_{\rho_e}, f_{\nu_0}\}$ adhere to the following relations

$$P(s_{i+1} = s'|s_i = s, a_i = a, x) = P(f_{s_i}(s, a, x, U))$$
$$P(r_i = r|s_i = s, a_i = a, x) = \delta(f_{r_i}(s, a, x) = r)$$
$$\pi^*(a_i = a|s_i = s, x) = P(f_{a_i}(a, s, x, U)$$
$$\rho_e(x) = P(f_{\rho_e}(x, U))$$
$$\nu_0(s_0|x) = P(f_{\nu_0}(s_0, x, U)),$$

where $\delta(\cdot)$ indicates the Dirac delta distribution.

We are now ready to define the confounded imitation problem. We define the (non-atomic) intervention $\sigma_x = \mathbf{do}(f_{\rho_e} \leftarrow f_{\rho_o})$ which replaces $f_{\rho_e}$ with $f_{\rho_o}$ in the contextual MDP SCM defined above. The goal of imitation learning is then to identify the quantities

$$P(a_i|s_i, x, \sigma_x = \mathbf{do}(f_{\rho_e} \leftarrow f_{\rho_o})) \quad , 0 \leq i \leq H - 1, \tag{6}$$

where, importantly, we assume we *only* have access to $P(s_i, a_i)$, $P(s_{i+1}|s_i, a_i, x)$, $P(s_0|x)$, and $P(x|\sigma_x = \mathbf{do}(f_{\rho_e} \leftarrow f_{\rho_o}))$. Notice that in our setting, $P(s_{i+1}|s_i, a_i, x)$, $P(s_0|x)$, and $P(x|\sigma_x = \mathbf{do}(f_{\rho_e} \leftarrow f_{\rho_o}))$ correspond to known quantities of the online environment, whereas $P(s_i, a_i)$ corresponds to the (partially observed) offline expert data. We also emphasize that $P(s_{i+1}|s_i, a_i, x)$ is not dependent on the intervention $\sigma_Z$. That is,

$$P(s_{i+1}|s_i, a_i, x, \sigma_x = \mathbf{do}(f_{\rho_e} \leftarrow f_{\rho_o})) = P(s_{i+1}|s_i, a_i, x).$$

**Remark.** *Our work studies a slightly different version of the identifiability problem in Equation (6), as we only wish to identify an optimal policy from the set $\Pi_{\mathcal{M}}^*$, as opposed to the single specific policy $\pi^*$. This requirement can be formalized by defining an extended SCM which includes all optimal policies in $\Pi_{\mathcal{M}}^*$, with the assumption that only one is observed (corresponding to the expert data).*

---

**Algorithm 4** RL using Expert Data with Unobserved Confounders (Complete Algorithm)

---

1: **input:** Expert data with missing context $\mathcal{D}^*, \lambda, \alpha, B, N, M > 0$, policy optimization algorithm `ALG-RL`
2: **init:** Policy $\pi^0$, global bonus reward network $g_\theta^*$
3: **for** $k = 1, \dots$ **do**
4:     Generate dataset of rollouts $\mathcal{R}_k \sim d_{\rho_o}^{\pi_{k-1}}(s, a)$
5:     Initialize local networks $g_{\theta_m}^m \leftarrow g_\theta, m \in [M]$
6:     **for** $m = 1, \dots M$ **do**
7:         Sample weight vector $w_m$ uniformly from $\Delta_n$
8:         **for** $e = 1 \dots N$ **do**
9:             Sample batch uniformly from $\mathcal{R}_k$, i.e., $\{s_i, a_i\}_{i=1}^B \overset{U}{\sim} \mathcal{R}_k$
10:            Sample batch according to weights $w_m$ from $\mathcal{D}^*$, i.e., $\{s_i^e, a_i^e\}_{i=1}^B \overset{w_m}{\sim} \mathcal{D}^*$
11:            Update $g_{\theta_m}^m$ according to

$$\nabla_{\theta_m} L_m(\theta_m) = \frac{1}{B} \sum_{i=1}^B \nabla_{\theta_m} \left[ (1-\alpha) f^*(g_{\theta_m}^m(s_i^e, a_i^e)) + \alpha f^*(g_{\theta_m}^m(s_i, a_i)) - g_{\theta_m}^m(s_i, a_i) \right]$$

12:         **end for**
13:         $m^* \in \arg\min_{m \in [M]} L_m(\theta_m)$
14:         Update global parameters from the selected local network $g_\theta^* \leftarrow g_{\theta_{m^*}}^{m^*}$
15:         $\pi^k \leftarrow$ `ALG-RL`$(r(s, a, x) - \lambda g_\theta^*(s, a))$
16:     **end for**
17: **end for**

---

| Name | Value | Comments |
|------|-------|----------|
| Batch size | 128 | |
| Learning rate | 5e−5 | |
| Rollout size | 19, 200 | |
| Total timesteps | 5e6 | |
| Num epochs | 50 | How many training epochs to do after each rollout |
| $\gamma$ | 0.95 | Discount factor |
| kl coef | 0.2 | Initial coefficient for KL divergence |
| kl target | 0.01 | Target value for KL divergence |
| GAE $\lambda$ | 1 | The GAE (lambda) parameter |
| Num workers | 40 | |

Table 2: Hyper-parameters used to train the PPO agent.

# E    IMPLEMENTATION DETAILS

Our experiments were based off of the recently proposed assistive-gym (Erickson et al., 2020) and recsim (Ie et al., 2019) environments. In this section we discuss further implementation details, hyperparameters, context distributions, and generation of the expert data.

**Algorithm Details.** A complete description of Algorithm 2 is presented in Algorithm 4. Specific hyperparameters used are shown in Tables 2 and 3. We implemented the algorithm using the RLlib framework (Liang et al., 2018). We used PPO (Schulman et al., 2017) as our policy-optimization algorithm. All neural networks consisted of two-layer fully connected MLPs with 100 parameters in each layer. We used the same rollout buffer (of size 19200 samples) for both our PPO agent as well as our imitation module, which estimated the augmented reward.

| Name | Value | Comments |
|---|---|---|
| Batch size | 128 | |
| Learning rate | 1e−4 | |
| Imitation Method | $\chi$-divergence | |
| Num epochs | 50 | How many training epochs to do after each rollout |
| $\alpha$ | 0.9 | $D_f$ regularization coefficient |
| $M$ | 10000 | Budget for CTS optimizer |

Table 3: Hyper-parameters used for imitation and CTS.

Motivated by Kostrikov et al. (2019), we regularized the expert demonstrations with samples from $d^\pi$. Particularly, we let $\alpha \in (0, 1]$, such that $1 - \alpha$ corresponds to the probabiilty of sampling an expert example and $\alpha$ corresponds to the probability of sampling from the replay. This leads to minimizing an augmented version of the $f$-divergence which can be written as

$$\min_{g:\mathcal{S}\times\mathcal{A}\mapsto\mathbb{R}} \mathbb{E}_{s,a\sim d^\pi_{\rho_o}(s,a|x)}[g(s,a) - \alpha f^*(g(s,a))] - (1-\alpha)\mathbb{E}_{s,a\sim d^{\pi^*}_{\rho_e}(s,a)}[f^*(g(s,a))].$$

Our imitation module consisted of two networks $g_\theta$ and $h_\theta$ as proposed in Fu et al. (2018). The "done" signal was also added to the state for training the imitation module. For training CTS we used the Nevergrad optimization platform (Rapin & Teytaud, 2018) with a budget of 10000 and one worker. Here a copied version of the networks $g_\theta$ and $h_\theta$ were used to for initialization and then approximate the minimum $D_f$.

For choosing $\lambda$ we used an adaptive strategy which ensured $\lambda$ balanced the RL objective with the imitation objective. Specifically, we used the following tradeoff between reward $r$ and bonus $g$

$$(1 - \lambda_{\text{adap}})r(s,a) + \lambda_{\text{adap}}g(s,a),$$

where $\lambda_{\text{adap}} = \frac{r_{\text{mean}}}{r_{\text{mean}}+g_{\text{mean}}}$. Here $r_{\text{mean}}$ corresponds to the average reward in the replay buffer and $g_{\text{mean}}$ to the average bonus in the replay buffer. By averaging the two, we maintained a similar scale to effectively use the expert data in all the evaluated environments without optimizing for $\lambda$.

**Context Distribution.** For each environment we used a varying context distribution in the expert data, with increasing distance to that of the online environment. The context distribution for the RecSim environment is formally described in Section 5. For the assistive-gym environment the context was defined by the following features: gender, mass, radius, height, patient impairment, and patient preferences. The patient's mass, radius, and height distributions were dependent on gender. The patient's impairment was given by either limited movement, weakness, or tremor (with sporadic movement). Finally, the patient's preferences were affected by the velocity and pressure of touch forces applied by the robot. We used default average values that were provided with the simulator. Particularly, we used the following distributions for each feature

$$\text{gender} \sim \text{Bern}(p_{\text{male}})$$
$$\text{mass(gender)} \sim \mathcal{N}(\mu_{\text{mass}}(\text{gender}), \sigma^2_{\text{mass}})$$
$$\text{radius(gender)} \sim \mathcal{N}(\mu_{\text{radius}}(\text{gender}), \sigma^2_{\text{radius}})$$
$$\text{height(gender)} \sim \mathcal{N}(\mu_{\text{height}}(\text{gender}), \sigma^2_{\text{height}})$$
$$\text{velocity weight} \sim \text{Unif}([\ell_{\text{vel}}, u_{\text{vel}}])$$
$$\text{force nontarget weight} \sim \text{Unif}([\ell_{\text{target}}, u_{\text{target}}])$$
$$\text{high forces} \sim \text{Unif}([\ell_{\text{high forces}}, u_{\text{high forces}}])$$
$$\text{food hit weight} \sim \text{Unif}([\ell_{\text{hit}}, u_{\text{hit}}])$$
$$\text{food velocity weight} \sim \text{Unif}([\ell_{\text{food vel}}, u_{\text{food vel}}])$$
$$\text{high pressures weight} \sim \text{Unif}([\ell_{\text{high pressure}}, u_{\text{high pressure}}])$$
$$\text{impairment} \sim \text{Multinomial}(p_{\text{none}}, p_{\text{limits}}, p_{\text{weakness}}, p_{\text{tremor}}).$$

| Name | Value | Name | Value | Name | Value |
|------|-------|------|-------|------|-------|
| $p_{\text{male}}$ | 0.3 | $\ell_{\text{vel}}$ | 0.225 | $\ell_{\text{high pressure}}$ | 0.009 |
| $\mu_{\text{mass}}(\text{male})$ | 78.4 | $u_{\text{vel}}$ | 0.275 | $u_{\text{high pressure}}$ | 0.011 |
| $\mu_{\text{mass}}(\text{female})$ | 62.5 | $\ell_{\text{target}}$ | 0.009 | $p_{\text{none}}$ | 0.1 |
| $\sigma^2_{\text{mass}}$ | 10 | $u_{\text{target}}$ | 0.011 | $p_{\text{limits}}$ | 0.4 |
| $\mu_{\text{radius}}(\text{male})$ | 1 | $\ell_{\text{high forces}}$ | 0.045 | $p_{\text{weakness}}$ | 0.3 |
| $\mu_{\text{radius}}(\text{female})$ | 1 | $u_{\text{high forces}}$ | 0.055 | $p_{\text{tremor}}$ | 0.2 |
| $\sigma^2_{\text{radius}}$ | 0.1 | $\ell_{\text{hit}}$ | 0.9 | | |
| $\mu_{\text{height}}(\text{male})$ | 1 | $u_{\text{hit}}$ | 1.1 | | |
| $\mu_{\text{height}}(\text{female})$ | 1 | $\ell_{\text{food vel}}$ | 0.9 | | |
| $\sigma^2_{\text{height}}$ | 0.1 | $\ell_{\text{food vel}}$ | 1.1 | | |

Table 4: Parameters for context distribution used in assistive-gym

| Name | Value | Name | Value | Name | Value |
|------|-------|------|-------|------|-------|
| $p_{\text{male}}$ | 0.8 | $\ell_{\text{vel}}$ | 0.225 | $\ell_{\text{high pressure}}$ | 0.009 |
| $\mu_{\text{mass}}(\text{male})$ | 88.4 | $u_{\text{vel}}$ | 0.275 | $u_{\text{high pressure}}$ | 0.111 |
| $\mu_{\text{mass}}(\text{female})$ | 72.5 | $\ell_{\text{target}}$ | 0.007 | $p_{\text{none}}$ | 0.1 |
| $\sigma^2_{\text{mass}}$ | 20 | $u_{\text{target}}$ | 0.016 | $p_{\text{limits}}$ | 0.1 |
| $\mu_{\text{radius}}(\text{male})$ | 0.9 | $\ell_{\text{high forces}}$ | 0.035 | $p_{\text{weakness}}$ | 0.1 |
| $\mu_{\text{radius}}(\text{female})$ | 0.9 | $u_{\text{high forces}}$ | 0.06 | $p_{\text{tremor}}$ | 0.7 |
| $\sigma^2_{\text{radius}}$ | 0.2 | $\ell_{\text{hit}}$ | 0.4 | | |
| $\mu_{\text{height}}(\text{male})$ | 1.1 | $u_{\text{hit}}$ | 2.1 | | |
| $\mu_{\text{height}}(\text{female})$ | 1.1 | $\ell_{\text{food vel}}$ | 0.4 | | |
| $\sigma^2_{\text{height}}$ | 0.2 | $\ell_{\text{food vel}}$ | 2.1 | | |

Table 5: Parameters for one of the shifted context distribution used in assistive-gym

The values for each distribution are provided in Table 4. For setting covariate shift, we used a a set of distributions that were shifted w.r.t. the default context distribution. We then sampled a shifted distribution w.p. $\beta$ and the default distribution w.p. $1 - \beta$. That is, when $\beta = 1$, the user sampled a context only from the shifted distribution. Table 5 shows an example of one of the shifted distribution that were used.

**Expert Data Generation.** For the assistive-gym experiments we a dense reward function for generating the expert data and a sparse one for our experiments using the expert data. Specifically, the dense reward function used the environment's default reward function, defined by

$$w_1 \cdot \text{distance to goal} + w_2 \cdot \text{action} + w_3 \cdot \text{task specific reward} + w_4 \cdot \text{preference score},$$

where the preferences were weighted according to the context features. Specific weights are provided in the implementation of assistive-gym (Erickson et al., 2020). The sparse reward function did not use the distance to goal (i.e., $w_1 = 0$).

# F MISSING PROOFS

## F.1 PROOFS FOR SECTION 3

We begin by proving two auxilary lemmas.

**Lemma 1.** *Let $\pi_2 \in \Upsilon_{\pi_1}$. Then, $\Upsilon_{\pi_1} = \Upsilon_{\pi_2}$.*

*Proof.* We show that $\Upsilon_{\pi_1} \subseteq \Upsilon_{\pi_2}$ and $\Upsilon_{\pi_2} \subseteq \Upsilon_{\pi_1}$.

Let $\pi \in \Upsilon_{\pi_1}$, then $d^\pi(s,a) = d^{\pi_1}(s,a)$. By our assumption, $\pi_2 \in \Upsilon_{\pi_1}$, then $d^{\pi_2}(s,a) = d^{\pi_1}(s,a)$. Hence, $d^\pi(s,a) = d^{\pi_2}(s,a)$. That is, $\pi \in \Upsilon_{\pi_2}$. This proves $\Upsilon_{\pi_1} \subseteq \Upsilon_{\pi_2}$.

Similarly, let $\pi \in \Upsilon_{\pi_2}$, then $d^\pi(s,a) = d^{\pi_2}(s,a)$. By our assumption, $\pi_2 \in \Upsilon_{\pi_1}$, then $d^{\pi_2}(s,a) = d^{\pi_1}(s,a)$. Hence, $d^\pi(s,a) = d^{\pi_1}(s,a)$. That is, $\pi \in \Upsilon_{\pi_1}$. This proves $\Upsilon_{\pi_2} \subseteq \Upsilon_{\pi_1}$, completing the proof. $\qquad\square$

**Lemma 2.** *Let $\pi_0$ be a deterministic policy and let $\mathcal{M}_0 = (\mathcal{S}, \mathcal{A}, \mathcal{X}, P, r_0, \gamma)$ such that $r_0(s,a,x) = \mathbf{1}\{a = \pi_0(s,x)\}$. Then $\pi_0$ is the unique, optimal policy in $\mathcal{M}_0$.*

*Proof.* By definition of $\pi_0$ and $r_0$,

$$r_0(s, \pi_0(s,x), x) = 1, \forall s \in \mathcal{S}, x \in \mathcal{X}.$$

In particular, $\mathbb{E}_{\pi_0}[r_0(s_t, a_t, x)] = 1$. Then

$$V^*_{\mathcal{M}_0} \leq (1-\gamma) \sum_{t=0}^{\infty} \gamma^t = \mathbb{E}_{\pi_0}\left[(1-\gamma) \sum_{t=0}^{\infty} \gamma^t r_0(s_t, a_t, x)\right] = V^{\pi_0}_{\mathcal{M}_0}.$$

This proves $\pi_0$ is an optimal policy. To prove uniqueness, assume by contradiction there exists an optimal policy $\pi_1 \neq \pi_0$. Then,

$$V^{\pi_1} = \mathbb{E}_{s,a,x \sim d^{\pi_1}(s,a,x)}[\mathbf{1}\{a = \pi_0(s,x)\}] = \mathbb{E}_{s,x \sim d^{\pi_1}(s,x)}\left[\mathbb{E}_{a \sim \pi_1(\cdot|s,x)}[\mathbf{1}\{a = \pi_0(s,x)\}]\right] < 1 = V^{\pi_0}_{\mathcal{M}_0}.$$

In contradiction to $\pi_1$ is optimal. Then, $\pi_0$ is a unique optimal policy. $\qquad\square$

We are now ready to prove Theorem 1.

**Theorem 1.** *[Sufficiency of $\Upsilon_{\pi^*}$] Assume $\rho_e \equiv \rho_o$. Let $\pi^* \in \Pi^*_{\mathcal{M}}$ and let $\pi_0 \in \Upsilon_{\pi^*}$. Then, $\Upsilon_{\pi^*} = \Upsilon_{\pi_0}$. Moreover, if $\pi_0 \neq \pi^*$, then there exists $r_0$ such that $\pi_0 \in \Pi^*_{\mathcal{M}_0}$ but $\pi^* \notin \Pi^*_{\mathcal{M}_0}$, where $\mathcal{M}_0 = (\mathcal{S}, \mathcal{A}, \mathcal{X}, P, r_0, \rho_o, \nu, \gamma)$.*

*Proof.* Let $\pi^* \in \Pi^*_{\mathcal{M}}$ and let $\pi_0 \in \Upsilon_{\pi^*}$. By Lemma 1, as $\pi_0 \in \Upsilon_{\pi^*}$, it holds that $\Upsilon_{\pi^*} = \Upsilon_{\pi_0}$. Next, choosing $r_0(s,a,x) = \mathbf{1}\{a = \pi_0(s,x)\}$, by Lemma 2 we get that $\pi_0$ is an optimal policy in $\mathcal{M}_0$. This proves $\pi_0 \in \Pi^*_{\mathcal{M}_0}$. Finally, by Lemma 2, $\Pi^*_{\mathcal{M}_0} = \{\pi_0\}$, proving $\pi^* \notin \Pi^*_{\mathcal{M}_0}$ if and only if $\pi^* \neq \pi_0$. $\qquad\square$

**Proposition 2.** *Define the mean policy* $\bar{\pi}(a|s,x) = \frac{\sum_{\pi \in \Upsilon_{\pi^*}} d^\pi(s,a,x)}{\sum_{\pi \in \Upsilon_{\pi^*}} \sum_{a'} d^\pi(s,a',x)}$, *and denote* $\alpha^* = \frac{|\Pi^*_{\mathcal{M}} \cap \Upsilon_{\pi^*}|}{|\Upsilon_{\pi^*}|} \in [0,1]$. *Then,* $v_{\mathcal{M}}(\bar{\pi}) \geq \alpha^* v^* + (1 - \alpha^*) \min_{\pi \in \Upsilon_{\pi^*}} v_{\mathcal{M}}(\pi)$.

*Proof.* Let $\tilde{\pi}$ as defined. Then by linearity of expectation

$$V^{\tilde{\pi}}_{\mathcal{M}} = \mathbb{E}_{s,a,x \sim d^{\tilde{\pi}}}[r(s,a,x)] = \frac{1}{|\Upsilon_{\pi^*}|} \sum_{\pi \in \Upsilon_{\pi^*}} \mathbb{E}_{s,a,x \sim d^\pi}[r(s,a,x)] = \frac{1}{|\Upsilon_{\pi^*}|} \sum_{\pi \in \Upsilon_{\pi^*}} V^\pi_{\mathcal{M}}.$$

Denote $B^* = \Pi_{\mathcal{M}}^* \cap \Upsilon_{\pi^*}$, then

$$
\begin{aligned}
V_{\mathcal{M}}^{\tilde{\pi}} &= \frac{1}{|\Upsilon_{\pi^*}|} \sum_{\pi \in B^*} V_{\mathcal{M}}^{\pi} + \frac{1}{|\Upsilon_{\pi^*}|} \sum_{\Upsilon_{\pi^*} \backslash B^*} V_{\mathcal{M}}^{\pi} \\
&= \frac{|B^*|}{|\Upsilon_{\pi^*}|} V_{\mathcal{M}}^* + \frac{1}{|\Upsilon_{\pi^*}|} \sum_{\Upsilon_{\pi^*} \backslash B^*} V_{\mathcal{M}}^{\pi} \\
&\geq \frac{|B^*|}{|\Upsilon_{\pi^*}|} V_{\mathcal{M}}^* + \frac{|\Upsilon_{\pi^*} \backslash B^*|}{|\Upsilon_{\pi^*}|} \min_{\pi \in \Upsilon_{\pi^*} \backslash B^*} V_{\mathcal{M}}^{\pi} \\
&\geq \frac{|B^*|}{|\Upsilon_{\pi^*}|} V_{\mathcal{M}}^* + \frac{|\Upsilon_{\pi^*} \backslash B^*|}{|\Upsilon_{\pi^*}|} \min_{\pi \in \Upsilon_{\pi^*}} V_{\mathcal{M}}^{\pi},
\end{aligned}
$$

completing the proof. $\qquad\square$

**Theorem 2.** *[Catastrophic Imitation] Assume $|\mathcal{X}| \geq |\mathcal{A}|$, and $P(s'|s, a, x) = P(s'|s, a, x')$ for all $x, x' \in \mathcal{X}$. Then $\exists \pi_{e,1}, \pi_{e,2}$ s.t. $\{\pi_{e,1}, \pi_{e,2}\}$ are non-identifiable, catastrophic expert policies.*

*Proof.* We first sketch the proof for the special case $\mathcal{X} = \{x_0, x_1\}$, $\mathcal{A} = \{a_0, a_1\}$ and a singleton state space $\mathcal{S} = \{s_0\}$. The general proof follows similarly and is given below.

By letting $\pi_1, \pi_2$ be the determinisic policies which choose opposite actions at opposite contexts, i.e., $\pi_1(x_i) = a_i, \pi_2(x_i) = a_{1-i}$, we can choose $\rho_e(x) = d^*(\pi_1(x))$ and $\widetilde{\rho}_e(x) = d^*(\pi_2(x))$ which yield

$$
\begin{aligned}
d_{\rho_e}^{\pi_1}(a) &= \sum_{i=0}^{1} \rho_e(x_i) \mathbf{1}\{a = \pi_1(x_i)\} \\
&= \sum_{i=1}^{2} d^*(\pi_1(x_i)) \mathbf{1}\{a = \pi_1(x_i)\} \\
&= \sum_{i=1}^{k} d^*(a_i) \mathbf{1}\{a_i = a\} := d^*(a).
\end{aligned}
$$

Similarly, $d_{\widetilde{\rho}_e}^{\pi_2}(a) = d^*(a)$.

For the second part of the proof choose $r_1(a, x) = \mathbf{1}\{x = x_i, a = a_i\}$ and $r_2(a, x) = \mathbf{1}\{x = x_i, a = a_{1-i}\}$. Notice that $\pi_i$ is optimal for $r_i$ under any distribution of contexts, yet $\pi_i$ achieves zero reward for $r_{1-i}$.

We now provide a complete proof for the general case.

Let $\rho_o, d^*(a)$. Without loss of generality, let $\mathcal{X} = \{x_0, \ldots, x_m\}$, $\mathcal{A} = \{a_0, \ldots, a_k\}$ with $m \geq k$, and denote $\mathcal{X}_k = \{x_1, \ldots, x_k\} \subseteq \mathcal{X}$. By definition there exists an injective function from $\mathcal{A}$ into $\mathcal{X}$.

Define

$$
f(x) = \begin{cases} a_i & , x = x_i, i = 0, \ldots, k \\ a_0 & , \text{o.w.} \end{cases}
$$

$$
g(x) = \begin{cases} a_{i+1 \ (\text{mod } k)} & , x = x_i, i = 0, \ldots, k \\ a_0 & , \text{o.w.} \end{cases}
$$

Then we can select $\pi_1, \pi_2, \rho_e, \widetilde{\rho}_e$ as follows

$$
\pi_1(a|x) = \mathbf{1}\{a = f(x), x \in \mathcal{X}_k\} + \frac{1}{k+1} \mathbf{1}\{x \notin \mathcal{X}_k\}
$$

$$
\pi_2(a|x) = \mathbf{1}\{a = g(x), x \in \mathcal{X}_k\} + \frac{1}{k+1} \mathbf{1}\{x \notin \mathcal{X}_k\},
$$

and

$$
\rho_e(x) = d^*(f(x)) \mathbf{1}\{x \in \mathcal{X}_k\},
$$
$$
\widetilde{\rho}_e(x) = d^*(g(x)) \mathbf{1}\{x \in \mathcal{X}_k\}.
$$

We get that

$$
\begin{aligned}
d_{\rho_e}^{\pi_1}(a) &= \sum_{i=1}^{m} \rho_e(x_i)\pi_1(a|x_i) \\
&= \sum_{i=1}^{k} d^*(f(x_i))\mathbf{1}\{a = f(x_i)\} \\
&= \sum_{i=1}^{k} d^*(a_i)\mathbf{1}\{a_i = a\} = d^*(a).
\end{aligned}
$$

Similarly,

$$
\begin{aligned}
d_{\widetilde{\rho}_e}^{\pi_2}(a) &= \sum_{i=1}^{k} d^*(g(x_i))\mathbf{1}\{a = g(x_i)\} \\
&= \sum_{i=1}^{k} d^*(a_{i+1 \ (\mathrm{mod}\ k)})\mathbf{1}\{a_{i+1 \ (\mathrm{mod}\ k)} = a\} \\
&= \sum_{i=1}^{k} d^*(a_i)\mathbf{1}\{a_i = a\} = d^*(a).
\end{aligned}
$$

This proves the first part of the theorem. For the other parts, choose $r_1, r_2$ as follows

$$
r_1(a, x) = \mathbf{1}\{x = x_i, a = a_i, 0 \le i \le k\}
$$
$$
r_2(a, x) = \mathbf{1}\{x = x_i, a = a_{i+1 \ (\mathrm{mod}\ k)}, 0 \le i \le k\}.
$$

Then, by definition, for any $P(x)$ such that $\mathrm{Supp}(P) \cap \mathcal{X}_k \ne \emptyset$,

$$
\mathbb{E}_{x \sim P(x), a \sim \pi_1(\cdot|x)}[r_1(a, x)] = 1 = \max_{\pi \in \Pi} \mathbb{E}_{x \sim P(x), a \sim \pi(\cdot|x)}[r_1(a, x)],
$$
$$
\mathbb{E}_{x \sim P(x), a \sim \pi_1(\cdot|x)}[r_2(a, x)] = 0 = \min_{\pi \in \Pi} \mathbb{E}_{x \sim P(x), a \sim \pi(\cdot|x)}[r_2(a, x)].
$$

And similarly,

$$
\mathbb{E}_{x \sim P(x), a \sim \pi_2(\cdot|x)}[r_1(a, x)] = 0 = \min_{\pi \in \Pi} \mathbb{E}_{x \sim P(x), a \sim \pi(\cdot|x)}[r_1(a, x)],
$$
$$
\mathbb{E}_{x \sim P(x), a \sim \pi_2(\cdot|x)}[r_2(a, x)] = 1 = \max_{\pi \in \Pi} \mathbb{E}_{x \sim P(x), a \sim \pi(\cdot|x)}[r_2(a, x)].
$$

The condition on the support holds for $\rho_e, \widetilde{\rho}_e$ by definition. If, $\mathrm{Supp}(\rho_o) \cap \mathcal{X}_k = \emptyset$, then the result holds trivially as $\mathbb{E}_{x \sim \rho_o(x), a \sim \pi(\cdot|x)}[r_1(a, x)] = \mathbb{E}_{x \sim \rho_o(x), a \sim \pi(\cdot|x)}[r_2(a, x)] = 0$ for all $\pi \in \Pi$. This completes the proof. $\qquad\square$

**Lemma 3.** *Assume $\mathrm{Supp}(\rho_o) \subseteq \mathrm{Supp}(\rho_e)$. Then*

$$
\arg\max_{\pi} \mathbb{E}_{x \sim \rho_e(x), s, a \sim d^\pi(s, a|x)}[r(s, a, x)] \subseteq \arg\max_{\pi} \mathbb{E}_{x \sim \rho_o(x), s, a \sim d^\pi(s, a|x)}[r(s, a, x)]
$$

*Proof.* For clarity we denote

$$
\begin{aligned}
\Pi_{\rho_e}^* &= \arg\max_{\pi} \mathbb{E}_{x \sim \rho_e(x), s, a \sim d^\pi(s, a|x)}[r(s, a, x)] \\
\Pi_{\rho_o}^* &= \arg\max_{\pi} \mathbb{E}_{x \sim \rho_o(x), s, a \sim d^\pi(s, a|x)}[r(s, a, x)] \\
\Pi_{\mathrm{Supp}(\rho_e)}^* &= \bigtimes_{x \in \mathrm{Supp}(\rho_e)} \arg\max_{\pi} \mathbb{E}_{s, a \sim d^\pi(s, a|x)}[r(s, a, x)].
\end{aligned}
$$

To prove the lemma, we will show $\Pi_{\rho_e}^* = \Pi_{\mathrm{Supp}(\rho_e)}^* \subseteq \Pi_{\rho_o}^*$.

We begin by proving $\Pi_{\rho_e}^* = \Pi_{\mathrm{Supp}(\rho_e)}^*$. Indeed, let $\pi^* \in \Pi_{\mathrm{Supp}(\rho_e)}^*$. Then, for any $x \in \mathrm{Supp}(\rho_e)$

$$
\mathbb{E}_{s, a \sim d^{\pi^*}(s, a|x)}[r(s, a, x)] = \max_{\pi} \mathbb{E}_{s, a \sim d^\pi(s, a|x)}[r(s, a, x)].
$$

In particular,

$$\mathbb{E}_{x \sim \rho_e(x), s, a \sim d^{\pi^*}(s, a|x)}[r(s, a, x)] = \mathbb{E}_{x \sim \rho_e(x)}\left[\max_\pi \mathbb{E}_{s, a \sim d^\pi(s, a|x)}[r(s, a, x)]\right] \geq \max_\pi \mathbb{E}_{x \sim \rho_e(x), s, a \sim d^{\pi^*}(s, a|x)}[r(s, a, x)],$$

where we used Jensen's inequality. This proves $\Pi^*_{\text{Supp}(\rho_e)} \subseteq \Pi^*_{\rho_e}$.

To see the other direction, let $\pi_e \in \Pi^*_{\rho_e}$ and assume by contradiction that $\pi_e \notin \Pi^*_{\text{Supp}(\rho_e)}$. Then, there exists $\tilde{x} \in \text{Supp}(\rho_e)$ such that

$$\mathbb{E}_{s, a \sim d^{\pi_e}(s, a|\tilde{x})}[r(s, a, \tilde{x})] < \max_\pi \mathbb{E}_{s, a \sim d^\pi(s, a|\tilde{x})}[r(s, a, \tilde{x})].$$

Define

$$\tilde{\pi}(\cdot|s, x) = \mathbf{1}\{x = \tilde{x}\}\pi_{\tilde{x}}(\cdot|s, \tilde{x}) + \mathbf{1}\{x \neq \tilde{x}\}\pi_e(\cdot|s, x),$$

where $\pi_{\tilde{x}} \in \arg\max_\pi \mathbb{E}_{s, a \sim d^\pi(s, a|\tilde{x})}[r(s, a, \tilde{x})]$. Then,

$$v(\pi_e) = P(x = \tilde{x})\mathbb{E}_{s, a \sim d^{\pi_e}(s, a|\tilde{x})}[r(s, a, \tilde{x})] + \sum_{x \in \text{Supp}(\rho_e) \setminus \{\tilde{x}\}} P(x)\mathbb{E}_{s, a \sim d^{\pi_e}(s, a|x)}[r(s, a, x)]$$

$$< P(x = \tilde{x})\mathbb{E}_{s, a \sim d^{\tilde{\pi}}(s, a|\tilde{x})}[r(s, a, \tilde{x})] + \sum_{x \in \text{Supp}(\rho_e) \setminus \{\tilde{x}\}} P(x)\mathbb{E}_{s, a \sim d^{\pi_e}(s, a|x)}[r(s, a, x)] = v(\tilde{\pi}),$$

in contradiction to $\pi_e \in \Pi^*_{\rho_e}$. This proves $\Pi^*_{\rho_e} \subseteq \Pi^*_{\text{Supp}(\rho_e)}$. We have thus shown that $\Pi^*_{\rho_e} = \Pi^*_{\text{Supp}(\rho_e)}$.

Finally, it is left to show that $\Pi^*_{\text{Supp}(\rho_e)} \subseteq \Pi^*_{\rho_o}$. Similar to before, let $\pi^* \in \Pi^*_{\text{Supp}(\rho_e)}$. Then, for any $x \in \text{Supp}(\rho_e)$, by Jensen's inequality

$$\mathbb{E}_{x \sim \rho_o(x), s, a \sim d^{\pi^*}(s, a|x)}[r(s, a, x)] = \mathbb{E}_{x \sim \rho_o(x)}\left[\max_\pi \mathbb{E}_{s, a \sim d^\pi(s, a|x)}[r(s, a, x)]\right] \geq \max_\pi \mathbb{E}_{x \sim \rho_o(x), s, a \sim d^{\pi^*}(s, a|x)}[r(s, a, x)].$$

This completes the proof. $\qquad\square$

**Theorem 3.** *[Sufficiency of Context-Free Reward]* Assume $\text{Supp}(\rho_o) \subseteq \text{Supp}(\rho_e)$ and $r(s, a, x) = r(s, a, x')$ for all $x, x' \in \mathcal{X}$. Then $\Upsilon_{\pi^*} \subseteq \Pi^*_\mathcal{M}$.

*Proof.* Let $\pi_0 \in \Upsilon_{\pi^*}$, we will show $\pi_0 \in \Pi^*_\mathcal{M}$. Since $r(s, a, x) = r(s, a, x')$ for all $x \in \mathcal{X}$ we denote $r(s, a) = r(s, a, x)$. By definition of $\Upsilon_{\pi^*}$ we have that.

$$d^{\pi_0}_{\rho_o}(s, a) = d^{\pi^*}_{\rho_e}(s, a)$$

Then,

$$v(\pi_0) = \mathbb{E}_{x \sim \rho_o(x), s, a \sim d^{\pi_0}(s, a|x)}[r(s, a)]$$

$$= \mathbb{E}_{x \sim \rho_o(x)}\left[\sum_{s \in \mathcal{S}, a \in \mathcal{A}} d^{\pi_0}(s, a \mid x)r(s, a)\right]$$

$$= \sum_{s \in \mathcal{S}, a \in \mathcal{A}} r(s, a)\mathbb{E}_{x \sim \rho_o(x)}[d^{\pi_0}(s, a \mid x)]$$

$$= \mathbb{E}_{s, a \sim d^{\pi_0}_{\rho_o}(s, a)}[r(s, a)]$$

$$= \mathbb{E}_{s, a \sim d^{\pi^*}_{\rho_e}(s, a)}[r(s, a)]$$

$$= \mathbb{E}_{x \sim \rho_e(x), s, a \sim d^{\pi^*}(s, a|x)}[r(s, a)]$$

$$= \max_\pi \mathbb{E}_{x \sim \rho_e(x), s, a \sim d^\pi(s, a|x)}[r(s, a)]$$

Then, $\pi_0 \in \arg\max_\pi \mathbb{E}_{x \sim \rho_e(x), s, a \sim d^\pi(s, a|x)}[r(s, a)]$. Applying Lemma 3

$$\pi_0 \in \arg\max_\pi \mathbb{E}_{x \sim \rho_o(x), s, a \sim d^\pi(s, a|x)}[r(s, a)] = \Pi^*_\mathcal{M},$$

completing the proof. $\qquad\square$

### F.2    PROOFS FOR SECTION 4

**Proposition 1.** *[Trajectory Sampling Equivalence] Let $\rho_s^*$ which minimizes Problem (P2) for some $\pi \in \Pi, g : \mathcal{S} \times \mathcal{A} \mapsto \mathbb{R}$, and assume $Supp(\rho_o) \subseteq Supp(\rho_e)$. Then, there exists $p^n \in \Delta_n$ such that $d_{\rho_s^*}^\pi(s, a) = \lim_{n \to \infty} \mathbb{E}_{i \sim p^n}\left[(1 - \gamma) \sum_{t=0}^\infty \gamma^t \mathbf{1}\{(s_t^i, a_t^i) = (s, a)\}\right].$*

*Proof.* We can write

$$d^\pi(s, a \mid x) = (1 - \gamma) \sum_{t=0}^\infty \gamma^t P(s_t = s, a_t = a | x)$$

$$= (1 - \gamma) \sum_\tau \sum_{t=0}^\infty \gamma^t P(s_t = s, a_t = a | x, \tau) P(\tau | x)$$

$$= (1 - \gamma) \sum_\tau \sum_{t=0}^\infty \gamma^t \mathbf{1}\{\tau_t = (s, a)\} P(\tau | x).$$

Then, denoting $P_{\rho_s^*}^\pi(\tau) = \mathbb{E}_{x \sim \rho_s^*}[P(\tau \mid x)]$, we get that

$$d_{\rho_s^*}^\pi(s, a) = (1 - \gamma) \sum_\tau \sum_{t=0}^\infty \gamma^t \mathbf{1}\{\tau_t = (s, a)\} P_{\rho_s^*}^\pi(\tau)$$

$$= \mathbb{E}_{\tau \sim P_{\rho_s^*}^\pi}\left[(1 - \gamma) \sum_{t=0}^\infty \gamma^t \mathbf{1}\{\tau_t = (s, a)\}\right].$$

Since, $Supp(\rho_o) \subseteq Supp(\rho_e)$, there exists $p^n \in \Delta_n$ such that $\mathbb{E}_{i \sim p^n}\left[(1 - \gamma) \sum_{t=0}^\infty \gamma^t \mathbf{1}\{s_t^i, a_t^i = (s, a)\}\right]$ is an unbiased estimator of $d_{\rho_s^*}^\pi(s, a)$. The result follows by the law of large numbers. □

**Theorem 5.** *Let* ALG-RL *be an approximate best response player that solves the RL problem in iteration $k$ to accuracy $\epsilon_k = \frac{1}{\sqrt{k}}$. Then, Algorithm 1 will converge to an $\epsilon$-optimal solution to Problem (P2) in $\mathcal{O}\left(\frac{1}{\epsilon^4}\right)$ samples.*

*Proof.* We begin by showing that $h(P) = \min_{x \in \Delta_n} D_f(P || \mathbb{E}_x[Q_x])$ is convex in $P$. We can write $D_f$ in its variational form, rewriting $h(P)$ as

$$h(P) = \min_{x \in \Delta_n} \max_{g : \mathcal{Z} \mapsto \mathbb{R}} \mathbb{E}_{z \sim P}[g(z)] - \mathbb{E}_{x, z \sim Q_x}[f^*(g(z))],$$

where

$$f^*(w) = \sup_y \{yw - f(y)\}.$$

We have that $\mathbb{E}_{z \sim P}[g(z)] - \mathbb{E}_{x, z \sim Q_x}[f^*(g(z))]$ is affine in $g$ and $x$. Therefore, strong duality holds, yielding

$$h(P) = \max_{g : \mathcal{Z} \mapsto \mathbb{R}} \min_{x \in \Delta_n} \mathbb{E}_{z \sim P}[g(z)] - \mathbb{E}_{x, z \sim Q_x}[f^*(g(z))]$$

$$= \max_{g : \mathcal{Z} \mapsto \mathbb{R}} \left\{ \mathbb{E}_{z \sim P}[g(z)] + \left( \max_{x \in \Delta_n} \mathbb{E}_{x, z \sim Q_x}[f^*(g(z))] \right) \right\}$$

We have that $\max_{x \in \Delta_n} \mathbb{E}_{x, z \sim Q_x}[f^*(g(z))]$ is convex in $g$ as a maximum over convex (affine) functions in a compact set. Therefore $h(P)$ is also convex as a maximum over convex functions.

Then, the objective in $Problem \ (P2)$ is convex in $d_{\rho_o}^\pi$. Following the meta algorithm framework for convex RL in Zahavy et al. (2021), we write the gradient of $D_f(d_{\rho_o}^\pi(s, a) || d_{\rho_e}^{\pi^*}(s, a))$. Notice that for any general $f$-divergence $D_f(x_i || y_i) = \mathbb{E}_{y_i}\left[ f\left(\frac{x_i}{y_i}\right) \right]$ it holds that

$$\nabla_{x_j} D_f(x_i || y_i) = 0, j \neq i,$$

and

$$\nabla_{x_i} D_f(x_i||y_i) = \nabla_{x_i} \mathbb{E}_{y_i}\left[f\left(\frac{x_i}{y_i}\right)\right] = \mathbb{E}_{y_i}\left[\frac{1}{y_i}\nabla_z f(z)\mid_{z=\frac{x_i}{y_i}}\right].$$

Specifically, for the $KL$-divergence, $D_{KL}(p_i||q_i) = -\mathbb{E}_{q_i}\left[\log\left(\frac{p_i}{q_i}\right)\right]$. Then,

$$\nabla_{p_i} D_{KL}(p_i||q_i) = \mathbb{E}_{q_i}\left[\frac{1}{p_i}\right].$$

Applying Lemma 2 of Zahavy et al. (2021) with a Follow the Leader (FTL) cost player completes the proof. $\qquad\square$

### F.3 PROOFS OF ADDITIONAL RESULTS IN APPENDIX

**Proposition 3.** *Assume $\rho_e \equiv \rho_o$ and $|\Upsilon_{\pi^*}| < \infty$. Then there exists $\lambda^* > 0$ such that for any $\lambda \in (0, \lambda^*)$, Algorithm 3 (with $\delta = 0$ sensitivity) will return $\bar{\pi}$ of Proposition 2 after exactly $|\Upsilon_{\pi^*}|$ iterations.*

*Proof.* Denote

$$\lambda_1^* = \max_{\pi \in \Pi_{\det}, \pi \notin \Upsilon_{\pi^*}, \pi' \in \Upsilon_{\pi^*}} d_{TV}\left(d_{\rho_o}^\pi(s,a,x), d_{\rho_o}^{\pi'}(s,a,x)\right),$$

$$\lambda_2^* = \min_{\pi \in \Pi_{\det}, \pi \notin \Upsilon_{\pi^*}} d_{TV}(d_{\rho_o}^\pi(s,a), d_{\rho_e}^{\pi^*}(s,a)),$$

where $d_{TV}$ is the total variation distance. Let $\lambda^* = \frac{\lambda_2^*}{\lambda_1^*}$ and $\lambda \in (0, \lambda^*)$ and notice that $\lambda^* > 0$.

To prove the result., we will show that at iteration $n$ of the algorithm $\pi_n \in \Upsilon_{\pi^*}$ and that either $\pi_n \notin \Upsilon_{n-1} := \{\pi_j\}_{j=1}^{n-1}$ or $\Upsilon_{n-1} = \Upsilon_{\pi^*}$.

**Base case ($n = 1$).** By the variational representation of the $f$-divergence,

$$\max_{g_0:\mathcal{S}\times\mathcal{A}\mapsto\mathbb{R}} \mathbb{E}_{s,a\sim d_{\rho_o}^\pi(s,a)}[g_0(s,a)] - \mathbb{E}_{s,a\sim d_{\rho_e}^{\pi^*}(s,a)}[f^*(g_0(s,a))] = d_{TV}(d_{\rho_o}^\pi(s,a), d_{\rho_e}^{\pi^*}(s,a)).$$

By definition $\Upsilon_{\pi^*} = \arg\min_{\pi \in \Pi_{\det}} d_{TV}(d_{\rho_o}^\pi(s,a)||d_{\rho_e}^{\pi^*}(s,a))$. Then, $\pi_1 \in \Upsilon_{\pi^*}$. Finally since $\Upsilon_0 = \emptyset$, we have that $\pi_1 \notin \Upsilon_0$.

**Induction step.** Suppose the claim holds for some $n = k$. We will show it holds for $n = k + 1$.

We begin by showing that $\pi_{k+1} \in \Upsilon_{\pi^*}$. Assume by contradiction that $\pi_{k+1} \in \Pi_{\det}, \pi_{k+1} \notin \Upsilon_{\pi^*}$. Using the variational form of the $f$-divergence,

$$\max_{g_i:\mathcal{S}\times\mathcal{A}\times\mathcal{X}} L_i(\pi_{k+1}; g_i) = d_{TV}(d_{\rho_o}^{\pi_{k+1}}(s,a,x), d_{\rho_o}^{\pi_i}(s,a,x)) \leq \lambda_1^*,$$

$$\max_{g_0:\mathcal{S}\times\mathcal{A}} L^*(\pi_{k+1}; g_0) = d_{TV}(d_{\rho_o}^{\pi_{k+1}}(s,a), d_{\rho_e}^{\pi^*}(s,a)) \geq \lambda_2^*.$$

We have that

$$\max_{\substack{g_0:\mathcal{S}\times\mathcal{A}\mapsto\mathbb{R}, \\ g_i:\mathcal{S}\times\mathcal{A}\times\mathcal{X}\mapsto\mathbb{R}}} L^*(\pi_{k+1}; g_0) - \lambda \min_i L_i(\pi_{k+1}; g_i) \geq \lambda_2^* - \lambda\lambda_1^* > \lambda_2^* - \lambda^*\lambda_1^* = 0.$$

Next, let $\tilde{\pi}_{k+1} \in \Upsilon_{\pi^*}$, then

$$\max_{\substack{g_0:\mathcal{S}\times\mathcal{A}\mapsto\mathbb{R}, \\ g_i:\mathcal{S}\times\mathcal{A}\times\mathcal{X}\mapsto\mathbb{R}}} L^*(\tilde{\pi}_{k+1}; g_0) - \lambda \min_i L_i(\tilde{\pi}_{k+1}; g_i) \leq 0,$$

where we used the fact that $L^*(\tilde{\pi}_{k+1}; g_0) = 0$ by definition of $\Upsilon_{\pi^*}$, and $L_i \geq 0$. We have reached a contradiction to $\pi_{k+1}$ being a solution to Equation (4). This proves that $\pi_{k+1} \in \Upsilon_{\pi^*}$.

Finally, we show that $\pi_{k+1} \notin \Upsilon_k$ if and only if $\Upsilon_k \neq \Upsilon_{\pi^*}$. First, notice that if $\Upsilon_k = \Upsilon_{\pi^*}$ then Equation (4) will return $\pi_{k+1} \in \Upsilon_k$ by definition of the total variation distance. Next, assume $\Upsilon_k \neq \Upsilon_{\pi^*}$ and assume by contradiction $\pi_{k+1} \in \Upsilon_k$. Then, $\exists i : \max_{g_i} L_i(\pi_{k+1}; g_i) = 0$, and $\max_{g_0:\mathcal{S}\times\mathcal{A}\mapsto\mathbb{R}} L^*(\pi_{k+1}; g_0) = 0$, by definition of $\Upsilon_{\pi^*}$. Hence,

$$\max_{\substack{g_0:\mathcal{S}\times\mathcal{A}\mapsto\mathbb{R}, \\ g_i:\mathcal{S}\times\mathcal{A}\times\mathcal{X}\mapsto\mathbb{R}}} L^*(\pi_{k+1}; g_0) - \lambda \min_i L_i(\pi_{k+1}; g_i) = 0.$$

In contrast, since $\Upsilon_k \neq \Upsilon_{\pi^*}$, there exists $\tilde{\pi} \in \Upsilon_{\pi^*}$ such that $\tilde{\pi} \notin \Upsilon_k$, and

$$\max_{\substack{g_0:\mathcal{S}\times\mathcal{A}\mapsto\mathbb{R}, \\ g_i:\mathcal{S}\times\mathcal{A}\times\mathcal{X}\mapsto\mathbb{R}}} L^*(\tilde{\pi}; g_0) - \lambda \min_i L_i(\tilde{\pi}; g_i) \leq \lambda_1^* < 0,$$

in contradiction to $\pi_{k+1}$ being a solution Equation (4). This completes the proof. $\qquad\square$

**Theorem 6.** *[Sufficiency of $\Upsilon_{\pi^*}^{\Gamma-1}$] Let Assumption 2 hold for some $\Gamma \geq 1$. Then $\pi^* \in \Upsilon_{\pi^*}^{\Gamma-1}$.*

*Proof.* Let $\pi \in \Pi$. We will show that $\pi \in \Upsilon_\pi^{\Gamma-1}$. By elementary algebra, we have that, under Assumption 2,

$$\rho_o(x)(1 - \Gamma^{-1}) + \Gamma^{-1} \leq \frac{\rho_o(x)}{\rho_e(x)} \leq \rho_o(x)(1 - \Gamma) + \Gamma.$$

Since $\text{Supp}(\rho_e) \subseteq \text{Supp}(\rho_o)$,

$$\begin{aligned}
d_{\rho_o}^\pi(s, a) &= \mathbb{E}_{x\sim\rho_o(x)}[d^\pi(s, a \mid x)] \\
&= \mathbb{E}_{x\sim\rho_e(x)}\left[\frac{\rho_o(x)}{\rho_e(x)} d^\pi(s, a \mid x)\right] \\
&\leq \mathbb{E}_{x\sim\rho_e(x)}[(\rho_o(x)(1 - \Gamma) + \Gamma) d^\pi(s, a \mid x)].
\end{aligned}$$

Subtracting $d_{\rho_e}^\pi$ from both sides we get that

$$\begin{aligned}
d_{\rho_o}^\pi(s, a) - d_{\rho_e}^\pi(s, a) &\leq \mathbb{E}_{x\sim\rho_e(x)}[(\rho_o(x)(1 - \Gamma) + \Gamma - 1) d^\pi(s, a \mid x)] \\
&= (\Gamma - 1)\mathbb{E}_{x\sim\rho_e(x)}[(1 - \rho_o(x)) d^\pi(s, a \mid x)] \\
&\leq \Gamma - 1.
\end{aligned}$$

Similarly,

$$d_{\rho_o}^\pi \geq \mathbb{E}_{x\sim\rho_e(x)}\left[\left(\rho_o(x)(1 - \Gamma^{-1}) + \Gamma^{-1}\right) d^\pi(s, a \mid x)\right].$$

Hence,

$$\begin{aligned}
d_{\rho_o}^\pi(s, a) - d_{\rho_e}^\pi(s, a) &\geq \mathbb{E}_{x\sim\rho_e(x)}\left[\left(\rho_o(x)(1 - \Gamma^{-1}) + \Gamma^{-1} - 1\right) d^\pi(s, a \mid x)\right] \\
&= \left(\Gamma^{-1} - 1\right)\mathbb{E}_{x\sim\rho_e(x)}[(1 - \rho_o(x)) d^\pi(s, a \mid x)] \\
&\geq -(1 - \Gamma^{-1}) \\
&\geq -(\Gamma - 1)
\end{aligned}$$

where the last two transitions hold since $\Gamma \geq 1$. Then, we have that

$$\left|d_{\rho_o}^\pi(s, a) - d_{\rho_e}^\pi(s, a)\right| \leq \Gamma - 1.$$

This completes the proof. $\qquad\square$

**Theorem 7.** *[Context Dependent Reward] Let $\epsilon : \mathcal{X} \mapsto \mathbb{R}$ of Definition 6. Denote $\epsilon_{oe} = \mathbb{E}_{x\sim\rho_o(x)}[\epsilon(x)] + \mathbb{E}_{x\sim\rho_e(x)}[\epsilon(x)]$. Then for any $\pi^* \in \Pi_\mathcal{M}^*$, $\pi_0 \in \Upsilon_{\pi^*}$*

$$v(\pi_0) \geq v(\pi^*) - \epsilon_{oe}.$$

*Proof.* Let $\pi^* \in \Pi_\mathcal{M}^*$, $\pi_0 \in \Upsilon_{\pi^*}$. The proof follows similar steps to that of Theorem 3.

$$v(\pi_0) = \mathbb{E}_{x \sim \rho_o(x), s, a \sim d^{\pi_0}(s,a|x)}[r(s, a, x)]$$

$$= \mathbb{E}_{x \sim \rho_o(x), s, a \sim d^{\pi_0}(s,a|x)}[r(s, a, x) - r_0(s, a)] + \mathbb{E}_{x \sim \rho_o(x), s, a \sim d^{\pi_0}(s,a|x)}[r_0(s, a)]$$

$$= \mathbb{E}_{x \sim \rho_o(x), s, a \sim d^{\pi_0}(s,a|x)}[r(s, a, x) - r_0(s, a)] + \mathbb{E}_{x \sim \rho_o(x)}\left[\sum_{s \in \mathcal{S}, a \in \mathcal{A}} d^{\pi_0}(s, a \mid x) r_0(s, a)\right]$$

$$= \mathbb{E}_{x \sim \rho_o(x), s, a \sim d^{\pi_0}(s,a|x)}[r(s, a, x) - r_0(s, a)] + \sum_{s \in \mathcal{S}, a \in \mathcal{A}} r_0(s, a) \mathbb{E}_{x \sim \rho_o(x)}[d^{\pi_0}(s, a \mid x)]$$

$$= \mathbb{E}_{x \sim \rho_o(x), s, a \sim d^{\pi_0}(s,a|x)}[r(s, a, x) - r_0(s, a)] + \mathbb{E}_{s, a \sim d_{\rho_o}^{\pi_0}(s,a)}[r_0(s, a)]$$

$$= \mathbb{E}_{x \sim \rho_o(x), s, a \sim d^{\pi_0}(s,a|x)}[r(s, a, x) - r_0(s, a)] + \mathbb{E}_{s, a \sim d_{\rho_e}^{\pi^*}(s,a)}[r_0(s, a)]$$

$$= \mathbb{E}_{x \sim \rho_o(x), s, a \sim d^{\pi_0}(s,a|x)}[r(s, a, x) - r_0(s, a)] + \mathbb{E}_{x \sim \rho_e(x), s, a \sim d^{\pi^*}(s,a|x)}[r_0(s, a)].$$

Then,

$$v(\pi^*) - v(\pi_0) = \mathbb{E}_{x \sim \rho_e(x), s, a \sim d^{\pi^*}(s,a|x)}[r(s, a, x) - r_0(s, a)] - \mathbb{E}_{x \sim \rho_o(x), s, a \sim d^{\pi_0}(s,a|x)}[r(s, a, x) - r_0(s, a)]$$

$$\leq \mathbb{E}_{x \sim \rho_e(x)}[\epsilon(x)] + \mathbb{E}_{x \sim \rho_o(x)}[\epsilon(x)],$$

completing the proof. $\square$

