# OpenReview forum: "On Covariate Shift of Latent Confounders in Imitation and Reinforcement Learning"
_ICLR.cc/2022/Conference — ICLR 2022 Poster_

### Official Review · Reviewer_JYzA · 2021-10-30

**Correctness:** 3
**Technical Novelty And Significance:** 2
**Empirical Novelty And Significance:** 2
**Recommendation:** 6
**Confidence:** 4

**Main Review:**

Strengths:

This paper tackles an important and relevant problem of RL in practice -- imitation learning with access to expert data that might be different from the actual experimental/online environment, and more importantly, that lacks contextual (confounders) information. Results in this paper cover settings that one may encounter in practice. Abstractly speaking, this problem is very similar to a problem that the statistics community starts to get interested in: that is, how to use data from different resources (e.g. a small trial data and a large, but potentially confounded observational data) to improve the qualify and confidence of decision making. Therefore, the authors should be applauded to make connections to causality by writing how their notation and setup can be translated into causal inference language in the appendix.

Proofs of the theoretical claims are not technically-dense and relatively self-complete, guarded by numerical experiments. The paper is quite well written and accessible to people who are not so familiar with RL but have backgrounds in a related subject (e.g. causality).

Weaknesses/minor comments:

The theoretical claims in this paper are all as expected and can be guessed by experts. Results in this paper are not a huge advancement of our understanding of RL but I do think this direction should be explored at some point. But as admitted by the authors, the lack of theoretical support that expert data improves learning efficiency in Section 4 is a main weakness of the paper, leaving a very important gap between theory and practice. Apart from this, I have the following minor comments.

1. In Appendix D, it might be helpful to further relate atomic/non-atomic interventions to terminologies like hard/soft interventions that the causal inference community is more familiar with.

2. The terminology "stationary distribution of a policy $\pi$" before Section 3 is confusing. It sounds like a probability distribution over policies. It might be helpful if the authors could choose a new terminology or make some further clarification when it was first introduced. For example, "stationary distribution of states and actions under a policy $\pi$'' might be a more accurate terminology.

3. Any figures in the paper should be self-contained. But one cannot get much information from reading Figure 2 and its legend alone. The authors should expand the legend a bit to explain the figure in more detail.

4. After Theorem 1, "the imitation solution is uniquely defined by the set $\Gamma_{\pi_{\ast}}$". Shouldn't this sentence be written as "the imitation solution is uniquely defined up to the set $\Gamma_{\pi_{\ast}}$"? Also "in selection of a suboptimal ..." should instead be "in the selection of a suboptimal ...".

5. Is it standard in the field of RL to assume experts data always follow the optimal policy? Could the authors provide some discussions on this assumption? To me, this assumption looks quite suspicious and might cast doubt on how the theoretical results and algorithms under this assumption could be useful to guide practice.



**Summary Of The Paper:**

This paper focuses on imitation learning with confounded expert data (with no contextual information) that might have covariate shifts. Theoretical results (upper and lower bounds) are obtained, supported by numerical experiments and real data applications. The main conclusions are: (1) with no access to reward in the online setting, it is impossible to learn the optimal policy with confounded expert data with covariate shifts; (2) but with access to reward, then it is possible to learn the optimal policy consistently.

**Summary Of The Review:**

Overall, this is a technically sound paper. Both the theoretical and empirical results in this paper are expected. But it is good to have a paper to confirm the related theory. Furthermore, considering that the authors did not manage to give a theorem on when expert data improves learning efficiency, I think it is fair to evaluate their contribution as marginally above the acceptance threshold but I would not fight for the acceptance of this paper at this stage.

---

> ### Author Response · Authors · 2021-11-17
> **Author Response**
>
> Thank you for your positive review and helpful comments. Please find our response below.
>
> Re terminology: As suggested, we’ve changed the phrase stationary distribution to its technically correct term: state-action frequency.
>
> Re Figure 2: As suggested, we added further details in the caption.
>
> Re Theorem 1: You are correct. We’ve rephrased this to avoid confusion.
>
> Re Expert Datasets: Shortly, very few papers have addressed the problem of imperfect imitation data [1]. Contemporary state of the art imitation algorithms do assume an optimal agent generated the data [2,3,4]. While non-optimal demonstrations is an interesting direction of research, it is orthogonal to our problem and is interesting to study this under the more basic non-confounding setting.
>
> Re sample complexity RL + imitation: We would like to address this point thoroughly as it seemed to have been the item that mostly concerned the reviewer. While we agree that a theoretical result on sample complexity of RL with confounded expert data is an important and interesting problem, we emphasize reasons why we believe such a result is not in the scope of our current paper:
>
> 1. We are not aware of any theoretical results on the sample complexity of imitation + RL - $\text{\emph{even in the unconfounded setting}}$. This is still an area of open research for $\text{\emph{non-confounded}}$ expert data. Our work focuses on the confounded setting, and we chose to empirically demonstrate the confounding effect on difficult benchmarks.
>
> 2. The assumptions needed to provide the requested theory go beyond the scope of the paper. Without further assumptions, one could construct an example for which confounded expert data (even without covariate shift) does not improve sample complexity. For instance, let the context space and action space be equal $\mathcal{X} = \mathcal{A}$, and let the optimal action be unique for every unique context. If the context distribution is uniform then the marginal action distribution will be uniform as well. In a linear bandit setting, this information will not assist in convergence in terms of regret, i.e., optimal regret would be $\mathcal{O}(d\sqrt{TK})$. There are various assumptions that could be made to achieve improved guarantees, such as minimal gap, or bounded confounding, and we believe that exploring them and their implications is far from trivial, and is best dealt with in future work.
>
> 3. We tested our algorithm on difficult benchmarks where a standard RL algorithm (without expert data) converges to a suboptimal solution. This allowed us to depict the effect of the degree of confounding. We believe our empirical results provide sufficient evidence for this effect, as well as our solution to overcome it.
>
> References: \
> [1] Gao, Yang, et al. "Reinforcement learning from imperfect demonstrations." arXiv preprint arXiv:1802.05313 (2018). \
> [2] Jonathan Ho and Stefano Ermon.  Generative adversarial imitation learning.Advances in neural information processing systems, 29:4565–4573, 2016. \
> [3] Justin Fu, Katie Luo, and Sergey Levine.   Learning robust rewards with adversarial inverse reinforcement learning.arXiv preprint arXiv:1710.11248, 2017. \
> [4] Ilya Kostrikov, Rob Fergus, Jonathan Tompson, and Ofir Nachum.  Offline reinforcement learning with fisher divergence critic regularization.  InInternational Conference on Machine Learning,pp. 5774–5783. PMLR, 2021

---

### Official Review · Reviewer_KtAT · 2021-10-31

**Correctness:** 3
**Technical Novelty And Significance:** 2
**Empirical Novelty And Significance:** 2
**Recommendation:** 5
**Confidence:** 4

**Main Review:**

**Regarding the ambiguity set (definition 1)**, it is claimed that this set consists of all deterministic policies that cannot be distinguished from $\pi^*$ based on the observed data. However, the condition only checks the marginal state/action distributions match that of the expert, but it should be possible that using the joint distributions along observed trajectories could reveal additional distinguishing information.

Consider the following example. Suppose we have horizon 2, two states: $s_0, s_1$, two actions: $a, b$, we always start at $s_0$, and transitions always go from $s_0 \to s_1$.
Suppose we have two contexts $x_1, x_2$  sampled uniformly and two policies $\pi_1, \pi_2$ that map contexts to action sequences as follows:

$\pi_1(x_1) = (a, a)$ (takes action a at timestep 0, and action a at timestep 1)

$\pi_1(x_2) = (b, b)$ (takes action b at timestep 0, and action b at timestep 1)

$\pi_2(x_1) = (a, b)$ (takes action a at timestep 0, and action b at timestep 1)

$\pi_2(x_2) = (b, a)$ (takes action b at timestep 0, and action a at timestep 1)

The state-action marginal for either policy is uniform over ((s0, a), (s0, b), (s1, a), (s1, b)), but the two are distinguishable if we look at the trajectories themselves instead of just state-action marginals.
As such, I don't believe the ambiguity set is actually the minimal set of candidate optimal policies, though I don't think this distinction is ultimately that important for the claims in the paper.

**Regarding proposition 1:** while the average policy is certainly better than the worst policy in the set, I disagree that this really means it is "robust" to the ambiguity set. Viewing \alpha* as simply the probability of getting an optimal policy if we selected a candidate policy at random, the inequality in prop 1 is (essentially) a statement that the mean policy has better value than uniformly guessing a candidate policy and running it (lower bounding the values of all non-optimal candidates with the value of the worst candidate), without quantifying how much better. I'm not convinced this is a very interesting result, and it's likely that stronger assumptions will be needed to obtain any stronger results.

The result in theorem 2 is unsurprising, but it is definitely good to formalize that imitating state-action marginals is insufficient and can be arbitrarily bad.

**Theorem 3:** The statement of theorem 3 intuitively makes a lot of sense, but doesn't appear to actually say that this makes the imitation problem actually easy, as no algorithm is given to actually find the candidate policies that are proven to be optimal in the theorem. Indeed, the definition of ambiguity sets might not even be reasonable when the context distribution shifts, as even applying the same expert policy with different context distribution would likely get different state-action marginals. If different context distributions induce different distributions over the initial state s_0, and supposing there's no way to return to s_0 after the intial timestep, then the ambiguity set can't contain any policies due to the context mismatch. In this case, the set of these candidate policies is actually empty, in which case theorem 3 is trivially true.

In **section 4**, the authors show that running RL regularized with the expert policy can lead to a suboptimal policy with context mismatch, and propose an algorithm to do RL regularized by the expert policy, using a learned context distribution to reweight the expert data to allow for consistency. While the proposed algorithm is interesting, the theoretical analysis is not very useful, since (as noted by the authors) it does not show that the expert data is actually useful for policy learning. The paper could be improved substantially by quantifying how the expert can improve the RL learning. For example, it would be good to establish sample complexity for the proposed algorithm in the cases where there actually is no covariate shift and with bounded cova
riate shift.

To summarize, I do not believe the theoretical results presented are currently strong contributions. I would particularly appreciate additional results on how the expert data is useful for the proposed RL algorithm, though the algorithm is potentially interesting even without any formal analysis. I suggest heavily reducing the pure imitation sections, as the sufficiency results appear to be uninteresting in those sections. I would suggest only leaving theorem 2 and discussing how matching the expert's state-action distribution is insufficient, as it directly motivates the modifications made in the proposed algorithm.

Regarding the actual proposed algorithm, I do believe it to be an interesting way to incoporate the expert data under covariate shift. Empirical results show that the proposed algorithm can outperform a baseline that directly uses the expert data distrib
ution without adaptively reweighting the context distribution. If the main issue with the naively imitating the expert is simply that it will not converge the optimal policy, I think it would also be important to compare to a slightly more complex baseline that simply places decaying weight on the expert imitaiton over time, allowing the policy to use the expert in initial exploration without being limited at convergence.
More generally, it would also be important to reference and compare to existing work on incorporating imperfect expert data in imitation and reinforcement learning settings (for example [1]).

I also have concerns over how applicable the proposed setting is. Is it a realistic assumption that our learned policy can observe the context, but we don't have access to it for the logged expert data? The experiments are synthetically constructed and don't provide a convincing example of why this setting is important. I would appreciate additional discussion on when the proposed setting is relevant.

[1] Gao, Yang, et al. "Reinforcement learning from imperfect demonstrations." arXiv preprint arXiv:1802.05313 (2018).


**Summary Of The Paper:**

The paper studies how to incoporate expert data with covariate shift, defined in a contextual MDP where expert data comes from a different context distribution and where the expert contexts are unobserved. The paper provides limited theoretical results for pure imitation based on state-action marginals only, in particular showing that matching state-aciton marginals alone can lead to arbitrarily bad policies with covariate shift in the contexts. They also propose a hybrid RL-imitation algorithm that utilizes the expert data without allowing it to bias the final solution.

**Summary Of The Review:**

I do not believe the paper is currently ready for acceptance. I believe the theoretical contributions are very limited. The proposed algorithm is potentially interesting, but I have concerns about the relevance of the proposed problem setting and how it compares to some simpler baselines. The paper would benefit from more thorough analysis (both theoretical and empirical) of the proposed algorithm as well as additional discussion about the problem setting.

---

> ### Author Response · Authors · 2021-11-17
> **Author Response**
>
> Thank you for your thorough review and your insightful comments. Please find our response below.
>
> Re ambiguity set: You are correct, we have uploaded a corrected version of the paper. As you’ve noted, this does not change any of the results in the paper. Also, notice that using the stationary distribution is a particular focus of our work due to the efficiency of sampling from it - an approach used in contemporary state of the art imitation algorithms [1,2,3].
>
> Re Proposition 1: We have moved this result to the appendix to make room for results on bounded confounding (from Appendix C). As for robustness, we have rephrased this part in the paper to avoid confusion.
>
> Re Theorem 3: When the set is not trivially empty (apart from arguably pathological cases) this result implies standard imitation learning techniques can be used to learn an optimal policy. A refined version of this result, as shown in the “bounded confounding” section, uses $\delta$-ambiguity sets to allow for divergence from the actual distribution. We find Theorems 2 and 3 to be essential for setting the ground to understanding when imitation learning can be easy and when it is fundamentally hard. We believe these are the two extremes of this spectrum. We have moved additional results on bounded imitation from the appendix to this section.
>
> Re related work: Thank you for this reference. This is indeed important and we’ve added additional discussion to the paper.
>
> Re sample complexity RL + imitation: We would like to address this point thoroughly as it seemed to have been the item that mostly concerned the reviewer. While we agree that a theoretical result on sample complexity of RL with confounded expert data is an important and interesting problem, we emphasize reasons why we believe such a result is not in the scope of our current paper:
>
> 1. We are not aware of any theoretical results on the sample complexity of imitation + RL - $\text{\emph{even in the unconfounded setting}}$. This is still an area of open research for $\text{\emph{non-confounded}}$ expert data. Our work focuses on the confounded setting, and we chose to empirically demonstrate the confounding effect on difficult benchmarks.
>
> 2. The assumptions needed to provide the requested theory go beyond the scope of the paper. Without further assumptions, one could construct an example for which confounded expert data (even without covariate shift) does not improve sample complexity. For instance, let the context space and action space be equal $\mathcal{X} = \mathcal{A}$, and let the optimal action be unique for every unique context. If the context distribution is uniform then the marginal action distribution will be uniform as well. In a linear bandit setting, this information will not assist in convergence in terms of regret, i.e., optimal regret would be $\mathcal{O}(d\sqrt{TK})$. There are various assumptions that could be made to achieve improved guarantees, such as minimal gap, or bounded confounding, and we believe that exploring them and their implications is far from trivial, and is best dealt with in future work.
>
> 3. We tested our algorithm on difficult benchmarks where a standard RL algorithm (without expert data) converges to a suboptimal solution. This allowed us to depict the effect of the degree of confounding. We believe our empirical results provide sufficient evidence for this effect, as well as our solution to overcome it.
>
> References: \
> [1] Jonathan Ho and Stefano Ermon.  Generative adversarial imitation learning.Advances in neural information processing systems, 29:4565–4573, 2016. \
> [2] Justin Fu, Katie Luo, and Sergey Levine.   Learning robust rewards with adversarial inverse reinforcement learning.arXiv preprint arXiv:1710.11248, 2017. \
> [3] Ilya Kostrikov, Rob Fergus, Jonathan Tompson, and Ofir Nachum.  Offline reinforcement learningwith fisher divergence critic regularization.  InInternational Conference on Machine Learning,pp. 5774–5783. PMLR, 2021

---

> > ### Author Response · Authors · 2021-11-17
> > **Author Response (Continued)**
> >
> > Re Setting: As the other reviewers agreed, the paper’s setting is practical and relevant to many real world scenarios. There are several examples for which this setting could occur. We enumerate some here and will add these as motivational discussion to the paper.
> >
> > 1. Human experts. Consider the problem of using expert human driver data to help create an autonomous vehicle. Data collected from human drivers does not contain all the observations the driver used to base her decision. These might include: weather, mood, specific types of pedestrians (e.g., angry, police), etc. In such scenarios, data collected at a certain time would have missing information in a later stage of development. For such a scenario, hidden covariates should be taken into account, particularly when distribution shift in the unobserved factors is present.
> >
> > 2. Privacy. It is very common, especially in the medical domain, that not all information can be provided (e.g., data from a hospital). Nevertheless, an online algorithm may have access to this information (e.g., in a specific local hospital) and want to use it to improve its policy.
> >
> > 3. Continuously changing features. In recommender systems, data collected on users changes over time. The full state of the user cannot be recorded since it may be extremely large or require expensive inquiries to the user. Nevertheless, when new features are added to the algorithm (e.g., due to a version update), the previous (old) data (for which such features were not recorded) would still be beneficial for imitation.

---

> > > ### Comment · Reviewer_KtAT · 2021-11-22
> > > **Concerns about problem setting are addressed**
> > >
> > > Thanks for adding additional discussion on the motivation of the problem setting. My concerns on this issue are addressed.

---

> > ### Comment · Reviewer_KtAT · 2021-11-22
> > **Still believe theorem 3 is not helpful, and comparisons to other work on imperfect experts are crucially still missing**
> >
> > **Updated summary**: I still believe the theoretical contributions in the paper are not meaningful, and so the proposed algorithm needs amore thorough empirical evaluation/comparisons to stand on its own.
> >
> > **Regarding theorem 3**: I would argue that the emptiness of the ambiguity set in theorem 3 is not at all pathological. Thinking about it more closely, I believe it is actually only *non-empty* in trivial or fairly contrived scenarios. The trivial scenario is if nothing in the MDP depend on the contexts, in which imitation obviously works since the context plays no role at all. Now if the dynamics do vary with the context $x$, any particular policy has an induced marginal state-action distribution for each particular context, which we denote as a vector $d^\pi_x$. The total marginal state-action distribution for a given context distribution is then just given by a weighted average of the per-context state-action distributions. For the expert policy to lie in its own ambiguity set with a changing context distribution, we must then have $\sum_{x\in \mathcal X} d^\pi_x \rho_e(x) = \sum_{x\in \mathcal X} d^\{\pi}_x \rho_o(x)$, which I believe to be a fairly contrived (and unlikely if the context affected the dynamics at all) condition. If the expert policy doesn't even lie in its own ambiguity set, this also doesn't bode well for the odds of finding another policy that can match the state-action marginals.
> >
> > Regardless of how strong of a statement theorem 3 is, I also believe it has littlebearing on the actual algorithm proposed, which is specifically designed to not constrain the algorithm to follow a suboptimal expert's state-action marginal. As such, I still believe that only keeping theorem 2 is sufficient context for the algorithm.
> >
> > **Regarding sample complexity of imitation**: I agree that theoretical results on the sample complexity of RL and imitation are likely beyond the reasonable scope of this paper. My point is more that the theory in this paper seems really overemphasized to me in the current state, as all of the theoretical results are straightforward/trivial, while adding the sample complexity results here would be the ideal way to have nontrivial theoretical contributions to this paper. That said, I don't believe having strong theoretical contributions is strictly necessary for this paper, provided there are strong empirical results with important baselines included.
> >
> > **Regarding related work**: I believe it is extremely important to have empirical comparisons against algorithms designed to for imperfect expert imitation (without specifically tackling the confounded problem setting), and not just discussing it in related work. I believe only having pure RL baselines to be insufficient for this problem.
> >
> > **Minor points:**
> > Theorem 1 now doesn't seem to offer any meaningful insight. It says the marginalized ambiguity set is sufficient, but nowhere does it actually say what it is suffices to do, especially since it offers no meaningful guarantees in performance. Again, I recommend dropping this section and jumping straight into the motivating failure case in Theorem 2.

---

> > > ### Author Response · Authors · 2021-11-22
> > > **Author Response**
> > >
> > > Thank you for your response. We appreciate your concerns. We strongly believe that our theoretical as well as empirical results may be interesting and even surprising to researchers in the field.
> > >
> > > **Re theorem 3:** In Appendix B.3 we provide an example for which Theorem 3 is interesting. In general, considering a grid-world setting in which the context accounts for the walls, a reward is provided only upon reaching the goal, and is thus independent on the placement of walls. We found it quite surprising that the agent does not require knowledge of the walls (which are randomly generated) in the data to correctly imitate the expert. We believe this environment suggests that such settings do exist and could be of interest. We will add a short description of this example to the paper, with reference to the appendix.
> > >
> > > We emphasize that the general idea of the two edge cases (context free reward vs context free transitions) are there to describe the two extremes of the problem. Removing this result may significantly hurt the clarity of the paper.
> > >
> > > **Re Theorem 1:** We will add clarification for the theorem to address your comment.
> > >
> > > **Re empirical results:** While we agree that our algorithm is an interesting approach for general imperfect expert data, it may distract the focus of our work. Since there doesn't exist prior work that empirically shows the issues of confounded expert data in high dimensional environments, we believe these results are important on their own right.
> > >
> > > Finally, following the reviewer's advice, we will run experiments for the proposed baseline (i.e., diminishing weight for the expert data) and add these to the final version of the paper.

---

### Official Review · Reviewer_BBsG · 2021-11-03

**Correctness:** 3
**Technical Novelty And Significance:** 3
**Empirical Novelty And Significance:** 3
**Recommendation:** 6
**Confidence:** 4

**Main Review:**

Strengths:
- This work tries to categorize the cases that using expert data is helpful in the problem of imitation learning in the presence of latent confounders. It provides cases that is infeasible to solve this problem from expert data. Moreover, it shows some cases this problem becomes feasible. Therefore, it is quite interesting to see the impossibility results.
- The authors proposed an algorithm in the case of the RL problem that can obtain an optimal policy in the case of covariate shift.

Weaknesses:
- The case of bounded confounding is more appealing and it makes sense to happen in practical settings. It is unclear why the authors put it in the appendix. Moreover, it is better to elaborate on Assumption 1 on page 16. How much is the set in Theorem 5 big?
- As mentioned by the authors, it is not determined whether using expert data improves the learning efficiency. It was shown only by some experiments. In fact, we know that we can converge to optimal policy without using expert data if we observe enough samples. Therefore, the advantage of using the proposed algorithm is not justified theoretically.

**Summary Of The Paper:**

The authors considered the problem of imitation and reinforcement learning in the setting of contextual MDP with latent confounders. They defined an ambiguity set, i.e., the set of all deterministic policies that match the marginalized stationary distributions of a given policy. In imitation learning, in the case of no covariate shift, no policy in the ambiguity set of optimal policy can be ruled out. Moreover, they showed that acting uniformly with respect to this set results in a policy that is better than the worst policy in the set. In the case of having covariate shift, the authors showed that imitating the policy of the expert might result in a catastrophic policy if the transition probabilities are context-free. However, if the reward function is independent of context, then the problem of imitation learning becomes feasible. Moreover, the case of bounded confounding is studied in the appendix. In particular, it is shown that under some conditions, the optimal policy is in a \delta-ambiguity set. Finally, in the problem of RL, the proposed algorithm converges to the optimal policy using corrective trajectory sampling.

**Summary Of The Review:**

In overall, the paper has some interesting results to characterize the cases that the problem of imitation learning becomes feasible. However, it seems that these are some sufficient conditions and it does not determine the boundaries of feasible cases. Moreover, in the case of the RL problem, the advantage of using the proposed algorithm is not studied theoretically.

---

> ### Author Response · Authors · 2021-11-17
> **Author Response**
>
> Thank you for your review and your helpful comments. Please find our response below.
>
> Re bounded confounding: We appreciate your advice and have moved some of the results on bounded confounding to the front matter.
>
> Re Assumption 1: This type of sensitivity assumption is common in causal inference literature [1,2,3], and is sometimes called the marginal sensitivity model [4]. The size of the set in Theorem 5 would largely depend on Γ and on the structure of the MDP.
>
> Re sample complexity RL + imitation: We would like to address this point thoroughly as it seemed to have been the item that mostly concerned the reviewer. While we agree that a theoretical result on sample complexity of RL with confounded expert data is an important and interesting problem, we emphasize reasons why we believe such a result is not in the scope of our current paper:
>
> 1. We are not aware of any theoretical results on the sample complexity of imitation + RL - $\text{\emph{even in the unconfounded setting}}$. This is still an area of open research for $\text{\emph{non-confounded}}$ expert data. Our work focuses on the confounded setting, and we chose to empirically demonstrate the confounding effect on difficult benchmarks.
>
> 2. The assumptions needed to provide the requested theory go beyond the scope of the paper. Without further assumptions, one could construct an example for which confounded expert data (even without covariate shift) does not improve sample complexity. For instance, let the context space and action space be equal $\mathcal{X} = \mathcal{A}$, and let the optimal action be unique for every unique context. If the context distribution is uniform then the marginal action distribution will be uniform as well. In a linear bandit setting, this information will not assist in convergence in terms of regret, i.e., optimal regret would be $\mathcal{O}(d\sqrt{TK})$. There are various assumptions that could be made to achieve improved guarantees, such as minimal gap, or bounded confounding, and we believe that exploring them and their implications is far from trivial, and is best dealt with in future work.
>
> 3. We tested our algorithm on difficult benchmarks where a standard RL algorithm (without expert data) converges to a suboptimal solution. This allowed us to depict the effect of the degree of confounding. We believe our empirical results provide sufficient evidence for this effect, as well as our solution to overcome it.
>
> References : \
> [1] Jesse Y Hsu and Dylan S Small.  Calibrating sensitivity analyses to observed covariates in observa-tional studies.Biometrics, 69(4):803–811, 2013. \
> [2] Hongseok  Namkoong,  Ramtin  Keramati,  Steve  Yadlowsky,  and  Emma  Brunskill.Off-policypolicy  evaluation  for  sequential  decisions  under  unobserved  confounding.arXiv  preprintarXiv:2003.05623, 2020. \
> [3] Nathan Kallus and Angela Zhou. Minimax-optimal policy learning under unobserved confounding.Management Science, 67(5):2870–2890, 2021. \
> [4] Tan, Z. A distributional approach for causal inference using propensity scores. Journal of the American Statistical Association, 101(476):1619–1637, 2006

---

> > ### Comment · Reviewer_BBsG · 2021-11-28
> > **Thanks for addressing my comments**
> >
> > Thanks for addressing my comments. I agree with the authors that analyzing the sample complexity might be out of the scope of this paper. I decided to keep my positive score.

---

### Official Review · Reviewer_26tA · 2021-11-03

**Correctness:** 3
**Technical Novelty And Significance:** 3
**Empirical Novelty And Significance:** 3
**Recommendation:** 6
**Confidence:** 3

**Main Review:**

The setting considered in this paper is interesting and practical, and the authors give a theoretical analysis about the limitation of imitation learning in such a setting. Also, the solutions given in the paper are simple and easy to follow. However, there are still some weak points in this paper:
- It would be better to show a causal diagram, which would be easier for readers to understand the problem definition.
- Considering that there are some methods try to solve the confounded imitation learning problem [1][2], it would be better to compare the performance of the proposed methods with theirs.
- The authors only perform experiments on assistant health/ Recommendation systems, which is less considered in the evaluation of RL methods. Can authors perform more experiments on robotics tasks? e.g. half-cheetah, hopper. As far as I know, there are some offline data is available to perform imitation learning.

[1] Kumor, D., Zhang, J., & Bareinboim, E. (2021). Sequential Causal Imitation Learning with Unobserved Confounders.

[2] Zhang, J., Kumor, D., & Bareinboim, E. (2020). Causal imitation learning with unobserved confounders. Advances in neural information processing systems, 33.

**Summary Of The Paper:**

This paper considers a very interesting setting:  the expert data in the imitation learning is confounded by the context, and the distribution of context may vary between expert data and online data. The authors give a analysis about the limitation of imitation learning methods under the counfounded imitation learning, and proposed a solution for it even if the counfounder distribution varies.

**Summary Of The Review:**

In summary, I think that the setting considered this paper is quite interesting, but it would be better to perform more experiments to evaluate the performance of the proposed method.

---

> ### Author Response · Authors · 2021-11-17
> **Author Response**
>
> Thank you for your positive review and your comments. Please find our response below.
>
> Re causal diagram: As suggested by the reviewer, we moved the causal diagram from the appendix to the front matter and referenced it in the preliminaries to make the preliminaries more clear.
>
> Re comparison to other methods: This is a very important point. We were inspired by [1], which is indeed closely related to our work (as we discuss in the related work section). Still, their algorithmic approach is tabular, and their work does not focus on the algorithmic aspect (e.g., complexity, scale). Hence, it is unclear how to implement their method in non-tabular settings with a long horizon. Moreover, our work is the first to propose a practical approach in this setting, building upon contemporary state of the art approaches for imitation learning [2,3,4].
>
> Re MuJoCo baselines: Thank you for your suggestion. Unfortunately not all standard RL baselines naturally fit our setting of hidden confounders. Even though expert datasets for MuJoCo exist (d4rl), fitting MuJoCo benchmarks to our setting requires extending the data with confounders practically defining a different benchmark. The empirical domains we have chosen have been meticulously selected to emphasize the effects of the confounded variables and the distribution shift on the optimized policy, as well as relate to the real-world motivation of the proposed setting.
>
> References: \
> [1] Kumor, Daniel, Junzhe Zhang, and Elias Bareinboim. "Sequential Causal Imitation Learning with Unobserved Confounders." (2021).
>  \
> [2] Jonathan Ho and Stefano Ermon.  Generative adversarial imitation learning.Advances in neural information processing systems, 29:4565–4573, 2016. \
> [3] Justin Fu, Katie Luo, and Sergey Levine.   Learning robust rewards with adversarial inverse reinforcement learning.arXiv preprint arXiv:1710.11248, 2017. \
> [4] Ilya Kostrikov, Rob Fergus, Jonathan Tompson, and Ofir Nachum.  Offline reinforcement learningwith fisher divergence critic regularization.  InInternational Conference on Machine Learning,pp. 5774–5783. PMLR, 2021

---

> > ### Comment · Reviewer_26tA · 2021-11-29
> > **Thanks for the authors's response**
> >
> > Thanks for the authors' response which addresses my concerns, so I keep my positive score.

---

### Author Response · Authors · 2021-11-17
**To All Reviewers**

We would like to thank the reviewers for reading our paper and for their insightful comments. We are pleased that the reviewers found our paper “interesting” (26tA, BBsG) and “easy to follow” (26tA) tackling “an important and relevant problem of RL in practice” (JYzA). We are also happy the reviewers acknowledged the importance of the connection we made between causal inference and reinforcement learning.

Despite the fact that imitation learning is a widely used approach, there is a dearth of theoretical results about the limits and possibilities of the approach. We view our work as a first step in mapping out the conditions under which confounded imitation learning is possible, and a demonstration of ways in which it can be made practical. We wish to point out that no work we are aware of has actually shown a positive result in the sense the reviewers mention - proving that expert data can improve standard RL ($\textbf{\emph{even in the unconfounded setting}}$), and that this is an important direction for future work. Indeed, even negative results are rare, as are practical methods of the type we present in our work.

---

### Decision · Program_Chairs · 2022-01-20

**Decision:**

Accept (Poster)

**Comment:**

The authors consider the problem of using expert data with unobserved confounders for
both imitation and reinforcement learning settings. They showed how latent confounders
negatively affect the learning process and proposed a sampling algorithm that mitigates the
impact and delivers good empirical results.

I agree with the reviewers, this is a borderline paper but with a preference to accept.
The most salient concern was the lack of clear contribution. While the algorithm is interesting
with good experimental results that attract interest, it lacks actual theoretical backbone.
That being said, the authors put in solid effort and addressed concerns sufficiently in the rebuttal stage.
Thus I would prefer to see it accepted. The proposed research direction should be explored in the future.